# Orbitofrontal cortex computes gaze-dependent comparisons between attributes rather than integrated values

Aster Q. Perkins[1], Erin L. Rich [1,2,3]*

1 Center for Neural Science, New York University, New York, New York, United States of America, 2 Nash Family Department of Neuroscience, Icahn School of Medicine at Mount Sinai, New York, New York, United States of America, 3 Friedman Brain Institute, Icahn School of Medicine at Mount Sinai, New York, New York, United States of America

* e.rich@nyu.edu

## Abstract

Economic decisions often require weighing multiple dimensions, or attributes. The orbitofrontal cortex (OFC) is thought to be important for computing the integrated value of an option from its attributes and comparing values to make a choice. Although OFC neurons are known to encode integrated values, evidence for value comparison has been limited. Here, we used a multi-attribute choice task for monkeys (*Macaca mulatta*) to investigate how OFC neurons integrate and compare multi-attribute options. By representing attributes with separate cues and using eye tracking to measure attention, we demonstrate that OFC neurons encode the value of attended attributes independent of other attributes in the same option. Encoding was negatively weighted by the value of the matching attribute in the other option, consistent with a comparison between like attributes. These results indicate that OFC computes comparisons between attributes rather than integrated values, and does so dynamically, shifting with the focus of attention.

## Introduction

We are routinely faced with choices that involve multiple dimensions, such as cost, risk, quality, or quantity. To make a decision, it is commonly thought that the brain first combines relevant features to compute the overall value, or utility, of each option, and then compares these values to arrive at a choice [1–5]. The orbitofrontal cortex (OFC) is critical for value-based decision-making [6–9], and has been proposed to carry out this process of value integration and comparison [3,10–12]. Consistent with this, neural activity in OFC correlates with integrated values [3,9–11,13–17], but so far there has been scant evidence for comparison of these values in OFC ([14,18–21] but see [9,22]). Another, often-overlooked feature of OFC is that many neurons encode the value of the individual dimensions, or attributes, of a choice

**Data availability statement:** Data and original analysis code are available at doi: https://doi.org/10.5061/dryad.f1vhhmh7t.

**Funding:** Funding support was provided by grant R01MH134845 to ELR, a fellowship from the Pew Scholars Program in Biomedical Sciences to ELR, and grant F31MH127901 to AQP. The funders had no role in study design, data collection and analysis, decision to publish, or preparation of the manuscript.

**Competing interests:** The authors have declared that no competing interests exist.

**Abbreviations:** AIC, Akaike information criterion; CPD, coefficient of partial determination; OFC, orbitofrontal cortex; BIC, Bayesian information criterion; GLM, Generalized linear model.

option [9,13–15,18,23,24]. These responses are understudied compared to the more prevalent integrated value signals, but the common assumption is that attribute values are encoded in order to compute the integrated value of an option which is then compared to other option values.

An alternate possibility is that attribute values are encoded because they are the decision variables that are compared. For instance, a choice might entail direct comparisons of cost versus cost or taste versus taste. Behavioral evidence supports this idea, showing that decision-makers often compare component attributes directly, either instead of or in addition to comparing integrated values [25–30]. Similarly, eye tracking in humans and monkeys has found that gaze frequently shifts between like attributes of two options, consistent with a decision process that involves comparison of similar attributes [30,31]. Shifts of visual attention not only provide insight into the decision process, but also affect value-related responses in OFC. For instance, in the absence of choice, directing attention to a reward-predicting cue amplifies OFC encoding of the cue's value [32,33]. Similarly, when choice options are presented sequentially, many neurons encode the value of the option that is currently presented, and therefore attended, rather than being selective for only the first or second option [9,18,19,22,34].

With this in mind, we aimed to test whether neurons in OFC encode and compare component attributes or integrated values of multi-attribute options, and how this is affected by natural shifts of visual attention as a decision is made. To do this, we designed a novel multi-attribute choice task for rhesus monkeys, in which attribute values were represented by physically separate visual cues that were presented simultaneously and varied independently. This design allowed us to use gaze as a measure of attention and test whether attending to an attribute impacts neural encoding of that attribute alone or the entire option to which it belongs, and which of these variables are used to compute comparisons. Consistent with previous reports, we found no evidence for comparison of integrated values. Instead, single neurons in OFC encoded the value of attended attributes independent of other attributes in the same option, and this encoding was negatively weighted by the value of the matching attribute in the other option, consistent with a comparison between like attributes. These results extend previous work showing that visual attention is a critical mediator of neural activity in OFC [22,32,33,35], and argue against the idea that OFC computes choices by comparing integrated option values. Instead, we identify a novel, attention-based mechanism of choice computation at the level of single neuron activity.

## Results

### Choice behavior indicates attribute-level comparisons

Two rhesus monkeys performed a multi-attribute decision-making task (Fig 1a). Choice options varied in sweetness (sucrose concentration) of a fluid reward and probability of delivery. The sweetness and probability of an option were each represented by a colored bar, whose height varied monotonically with attribute magnitude. Bar color indicated whether the mapping between height and value was direct (i.e.,

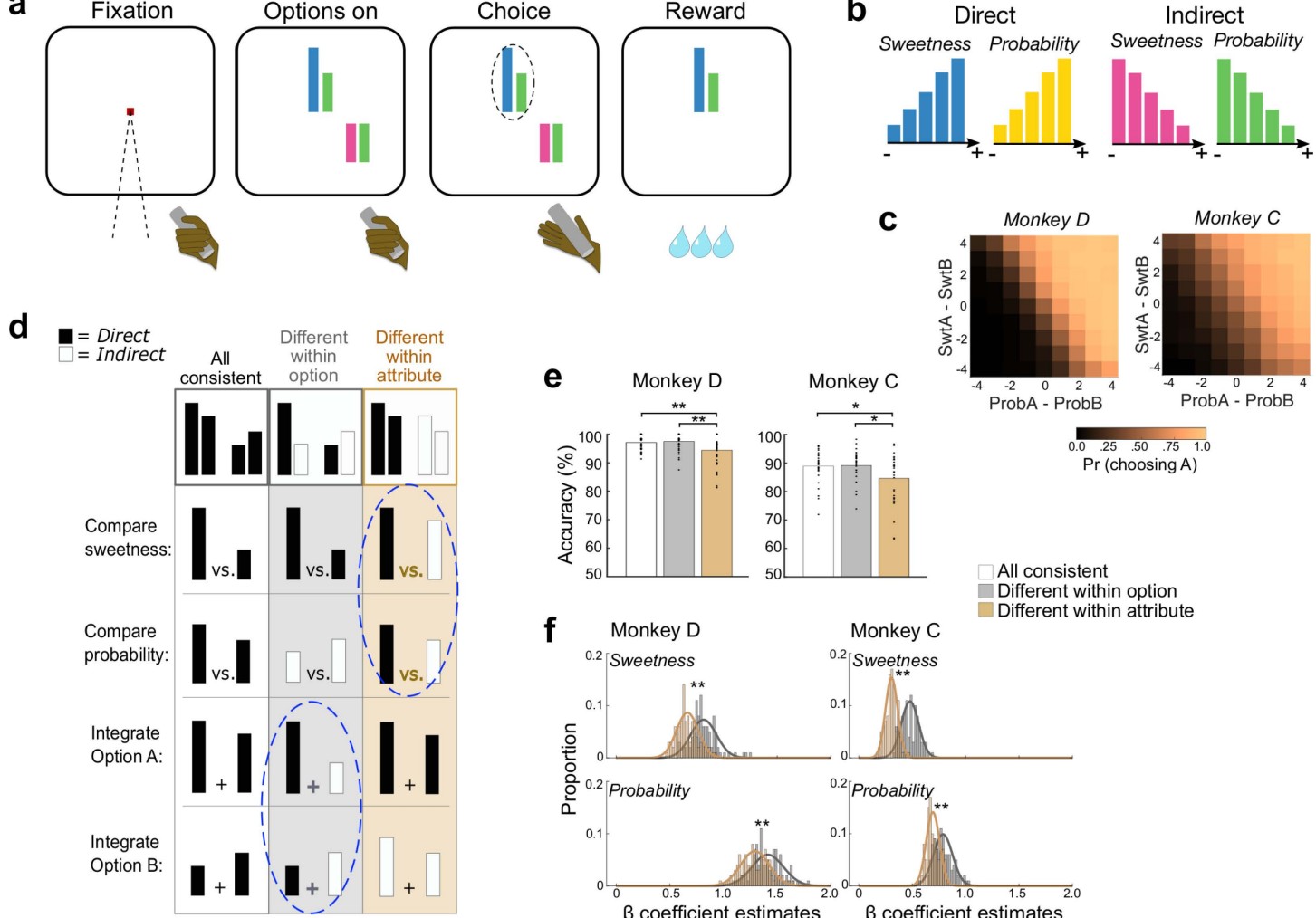

**Fig 1. Multi-attribute choice task. (a)** Left to right: Monkeys initiated a trial by holding a touch-sensitive bar and fixating a central point. Choice options were represented by a pair of colored bars of variable height. The left bar indicated the sweetness of an option and the right bar indicated the probability of receiving it. Monkeys freely viewed the options while holding the touch bar, and reported a choice by releasing the bar while directing gaze to one option. **(b)** Sweetness and probability attributes were represented by colored bars, whose height indicated the value of the attribute. Values varied monotonically in 5 increments. Blue and yellow bar heights directly mapped to the value of the attribute ($-$ = low, $+$ = high). Magenta and green bar heights indirectly mapped to the value of the attribute. **(c)** The probability of choosing option A (arbitrarily designated) increased with the difference between option A and B probabilities (ProbA $-$ ProbB) and sweetnesses (SwtA $-$ SwtB). Differences were computed from ordinal levels of each attribute. **(d)** Schematic of three trial types. Top row: Example decisions between two pairs of bars. White column: all attributes have the same mapping, either all direct (shown) or all indirect (not shown). All putative choice computations would take place between bars of the same mapping. Gray: mappings differ among attributes within an option, but like attributes are the same, so that attribute comparisons take place between bars with the same mapping, but integration takes place across different mappings (blue dashed circle). Orange: mappings differ within like attributes but are the same within an option, so that attribute comparisons take place between bars with different mappings (blue dashed circle), but value integration takes place between bars with the same mapping. **(e)** Accuracy on trials with an objectively better option is consistently lower when like attributes differ (orange). Bars show averages, points show session-wise accuracy in each trial type. Post-hoc comparisons *$p \leq 0.05$, **$p \leq 0.01$. **(f)** Weights for each attribute were estimated by beta coefficients from logistic regressions predicting choice probabilities separately from groups of trials in which attribute mapping differed within option (gray) or within attribute (orange) as in **d**. Since two coefficients of opposite sign were estimated for each attribute (for option A and B), plots show the mean absolute value of these. Histograms show bootstrapped sets of trials of each type. Lower coefficients indicate more choice variability. **Wilcoxon rank-sum test $p < 0.001$.

bigger is better) or indirect (bigger is worse). Sweeter and more probable outcomes were indicated by larger blue and yellow bars (direct mapping) or smaller magenta and green bars (indirect mapping) (Fig 1b).

Monkeys D and C performed 43 and 30 sessions, respectively, and averaged 1,085 and 1,203 trials per session. Choice probabilities indicated that they weighted both attributes to select sweeter, more probable options (Fig 1c). Foundational models of economic choice, such as expected utility theory [36] and prospect theory [37] assume that probability is multiplicatively combined with other attributes to calculate option values (but see [38,39]). In contrast, we previously found that behavior in this task is better fit by models that combine attributes linearly [31]. Formal model comparisons found that choice behavior in this data set was also better fit by an additive rather than multiplicative model (S1a Fig). Since these models are highly similar, this was confirmed with model recovery, showing that we can reliably select the correct generative model from simulated choices (S1b Fig). Therefore, choice behavior of both monkeys indicated that they computed choices by weighing and linearly combining sweetness and probability attributes, and we used this model for further analyses (S1c–S1j Fig).

Next, we separated trials with different attribute mappings to assess inefficiencies in the decision process [31]. We considered two operations that could be involved in computing decisions: integrating values within an option, and directly comparing like attributes of the different options (i.e., sweetness versus sweetness and probability versus probability). Integration or comparison should be easier when the relevant bars have the same mapping, since combining the area of two bars or deciding which of two bars is taller could be performed with simple perceptual judgements. However, when the relevant bars have different mappings (direct versus indirect), the monkeys must translate across mappings by referencing an internal representation of the bar's meaning – either its position on a value scale or its specific attribute magnitude (e.g., 50% probability) – in order to integrate or compare the information. This process can introduce subtle but measurable inefficiencies into the choice process [31]. To measure this, we separately analyzed behavior during three trial types. First were trials in which all four bars had a consistent mapping (all direct or all indirect, Fig 1d, white), so that integration of attributes within an option or comparison of like attributes across options would always rely on bars with the same mapping. We compared these to trials in which two bars were direct and two were indirect, in different arrangements. When mappings differed within option but like attributes had the same mapping (i.e., sweetness matched sweetness and probability matched probability), then attribute-level comparisons would take place between bars with the same mapping, but value integration would involve bars with different mappings (Fig 1d, gray). On the other hand, if the two attributes within each option had the same mapping but differed from the two attributes of the other option, then combining sweetness and probability into integrated values would involve bars of the same mapping, whereas comparing like attributes without integration would occur across different mappings (Fig 1d, orange). Therefore, separating trials according to whether mappings are mismatched within option or within attribute allowed us to test whether the need to translate between mappings impeded specific component processes that might underlie choice computation.

We first tested this without relying on behavior models by subselecting trials in which one option was superior to the other in both sweetness and probability, and therefore was objectively better ($n = 12,038$ and $8,888$ trials across all sessions, Monkey D and C, respectively). Focusing on these trials allowed us to compute choice accuracies without needing to account for subjective weighting of sweetness versus probability. In both subjects, accuracy on choices with an objectively better option was lower when mappings differed within like attributes, but was unaffected when mappings differed within an option (ANOVA Monkey D: $F_{2,126} = 9.32$, $p = 1.68 \times 10^{-4}$, Tukey-corrected post-hoc comparisons: all consistent versus differ within option $p = 0.86$, all consistent versus differ within attribute $p = 0.002$, differ within option versus differ within attribute $p = 2.13 \times 10^{-4}$; Monkey C: $F_{2,87} = 4.01$, $p = 0.02$, Tukey-corrected post-hoc comparisons: all consistent versus differ within option $p = 0.99$, all consistent versus differ within attribute $p = 0.050$, differ within option versus differ within attribute $p = 0.036$) (Fig 1e). Consistent with our previous findings [31], this pattern suggests that choices in this task involve, at least in part, comparisons between like attributes.

Next, we used the same attribute arrangements to test for attribute comparisons across all choices, not just those with an objectively better option. To do this, we separately modeled trials in which bar mappings differed within an option (Fig 1d, gray) or within like attributes (Fig 1d, orange) with logistic regressions (Eqs. 1, 3). If different bar mappings impeded within-attribute comparisons, we would expect less consistent choices on trials where mappings differed within like attributes (orange), quantified as shallower fitted slopes. Indeed, estimated beta coefficients were consistently lower on these trials, indicating more choice variability (Fig 1f). Taken together, patterns of choice behavior provide consistent evidence that attribute comparisons contribute to decisions in this task. Given this, our goal was to assess whether and how such comparisons are made by OFC neurons.

## OFC neurons encode attributes of multi-attribute options

We recorded a total of 322 well-isolated single neurons from OFC as monkeys performed the multi-attribute choice task (163 Monkey D, 159 Monkey C). To understand how neurons encode information about sweetness and probability, we analyzed each neuron's activity in sliding windows with a regression model predicting firing rate from the value of each attribute of the chosen and unchosen options, as well as the mappings (direct or indirect) of each attribute (Eq. 5). Neurons that encode the integrated value of a choice should have non-zero regression coefficients for both attributes of the chosen option, simultaneously with the same sign. Some neurons fit these criteria (Figs 2a and S2a), but this encoding pattern was relatively infrequent and accounted for approximately 10% of neurons at any time (Figs 2b and S2e). If the value of the chosen option was compared to the unchosen option, we would expect to see the two attributes of the unchosen option also jointly represented, but almost no neurons did this (Figs 2b and S2e). Instead, many neurons encoded the value of only one attribute, most frequently an attribute of the chosen option (Figs 2a and S2b). These neurons showed no tendency for firing rates to be modulated by the value of the paired attribute in the same option (S2f Fig and S2g Fig). Such single attribute responses were found in 20–30% of neurons in any time bin during the choice epoch, making them the most common type of response (Figs 2b and S2e).

We also found neurons that encoded the same attribute of both chosen and unchosen options, with firing rates modulated in the same direction (e.g., higher firing with higher value of either option's sweetness) (Figs 2a and S2c). This type of response is inconsistent with a comparison signal, but was found in approximately 15% of neurons, soon after the options appeared (Figs 2b and S2e). We reasoned that, if these neurons encode unintegrated attributes before a choice is determined, categorizing attributes according to which option was chosen or not chosen would be the wrong reference frame for quantifying these responses. Instead, the neurons might distinguish attributes according to whether they are *better* or *worse* than the other like attribute on that trial, regardless of which one belongs to the option that is ultimately chosen (Fig 2c). Therefore, we tested whether neurons encoded the values of the better or worse sweetness or probability (Eq. 6). Since better attributes tend to be chosen, this model is correlated with the previous one, so we used model comparisons to distinguish them.

We found that up to 20% of OFC neurons were best fit by the better/worse model, indicating that they encoded like attributes relative to one another, not relative to the choice (Fig 2d). This encoding was most frequent early in the choice, and about 500 ms later gave way to a predominant encoding of attributes according to whether they were chosen or not. This pattern suggests that OFC initially processes unintegrated attributes in a reference frame that does not depend on, and therefore can precede, a choice. We considered models in which attributes of an option are integrated in different ways, but the most common encoding in both subjects was of independent attributes, either as the better or worse sweetness/probability early in the trial, or as part of the chosen or unchosen option later in the trial (S2i Fig). There was a tendency for the same neurons to shift from coding attributes relative to one another (better/worse reference frame) to encoding them relative to the choice (chosen/unchosen reference frame) (binomial test, $p = 5.15 \times 10^{-4}$), although there were also many neurons that encoded chosen/unchosen attributes without previously encoding better/worse attributes (Fig 2e). This pattern was found in each monkey separately, but only significant in one (Monkey D, C: $p = 0.002$, $p = 0.056$).

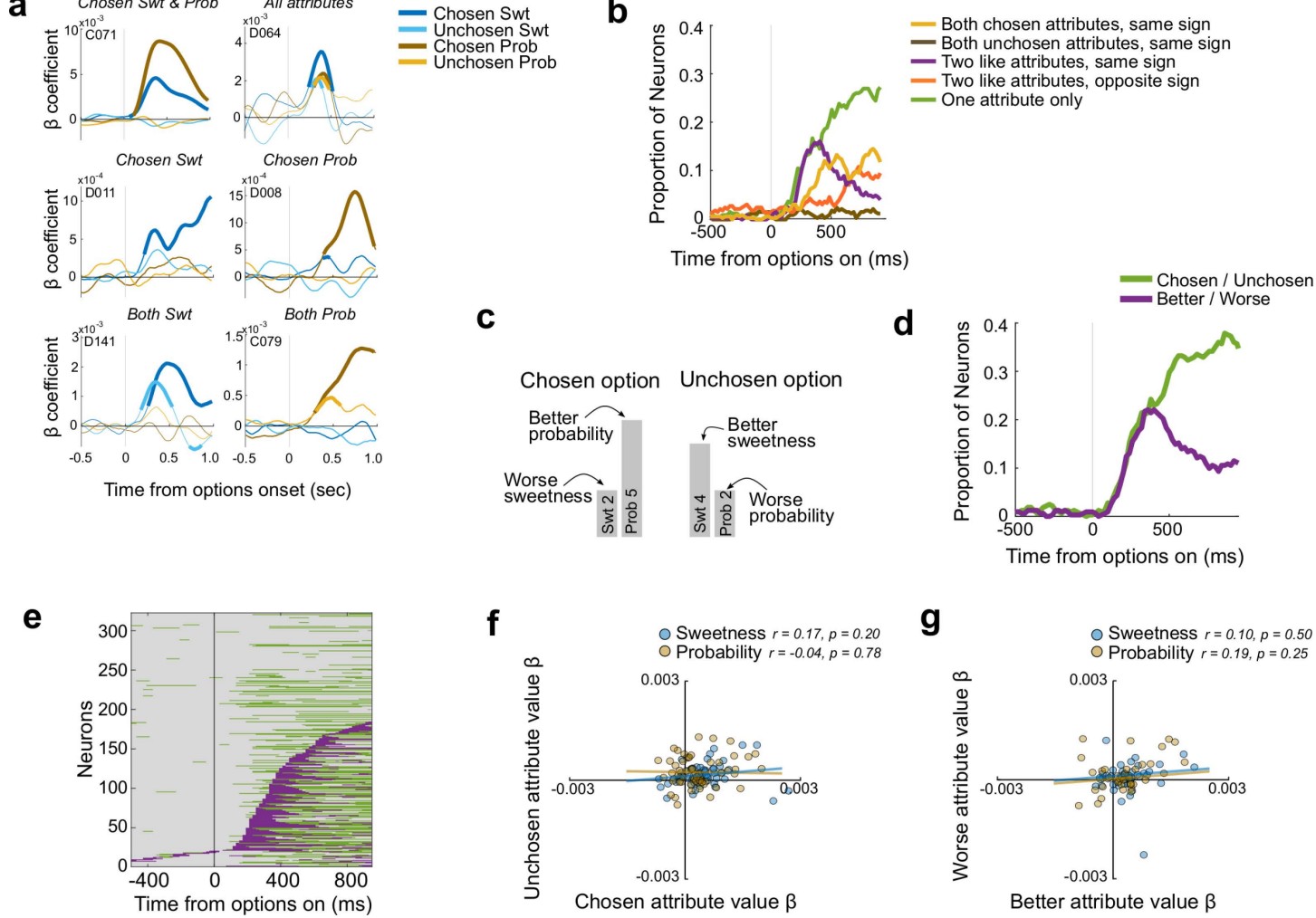

**Fig 2. Attributes encoded by OFC neurons. (a)** Types of attribute encoding in six example neurons. Lines show signed coefficient estimates for chosen or unchosen sweetness (Swt) or probability (Prob) over time. Thick segments denote regions of significance ($p \leq 0.01 \times 3$ consecutive time bins). Top row left to right: A neuron encoding both sweetness and probability of the chosen option. A neuron whose firing rate varies with all four attributes. Middle row: Neurons encoding the chosen sweetness only (left) or chosen probability only (right). Bottom row: Neurons whose firing rates vary with both chosen and unchosen sweetnesses (left) or probabilities (right). **(b)** Proportion of neurons with different encoding patterns. **(c)** Schematic showing relative value of an attribute that is discordant with the chosen options. The choice is between a low (level 2) sweetness and high (level 5) probability vs. a high (level 4) sweetness and low (level 2) probability, and the first option is selected. Only direct mappings are shown for simplicity. The chosen/unchosen model would assign attribute values for chosen sweetness, chosen probability, unchosen sweetness, unchosen probability as [2 5 4 2], but the better/worse model for the same trial would assign better sweetness, better probability, worse sweetness, worse probability as [4 5 2 2]. **(d)** Proportion of neurons assigned to each model. **(e)** Neurons assigned to the best/worst (purple) or chosen/unchosen (green) model over time, sorted by the time of better/worse encoding. Many neurons shifted from encoding better/worse attributes early and chosen/unchosen attributes later. **(f–g)** There were no correlations between beta coefficients for chosen and unchosen attributes or better and worse attributes.

So far, we found that many OFC neurons encode attributes separately. Early in the choice, they are encoded in relation to the same attribute in the other option (i.e., better/worse reference frame) and later in relation to the decision (i.e., chosen/unchosen reference frame). This could suggest a comparison process between better and worse attributes that produces a chosen option. Therefore, we tested for antagonistic value coding consistent with a comparison signal, by looking for inverse relationships among the regression coefficients. However, in both the better/worse reference frame

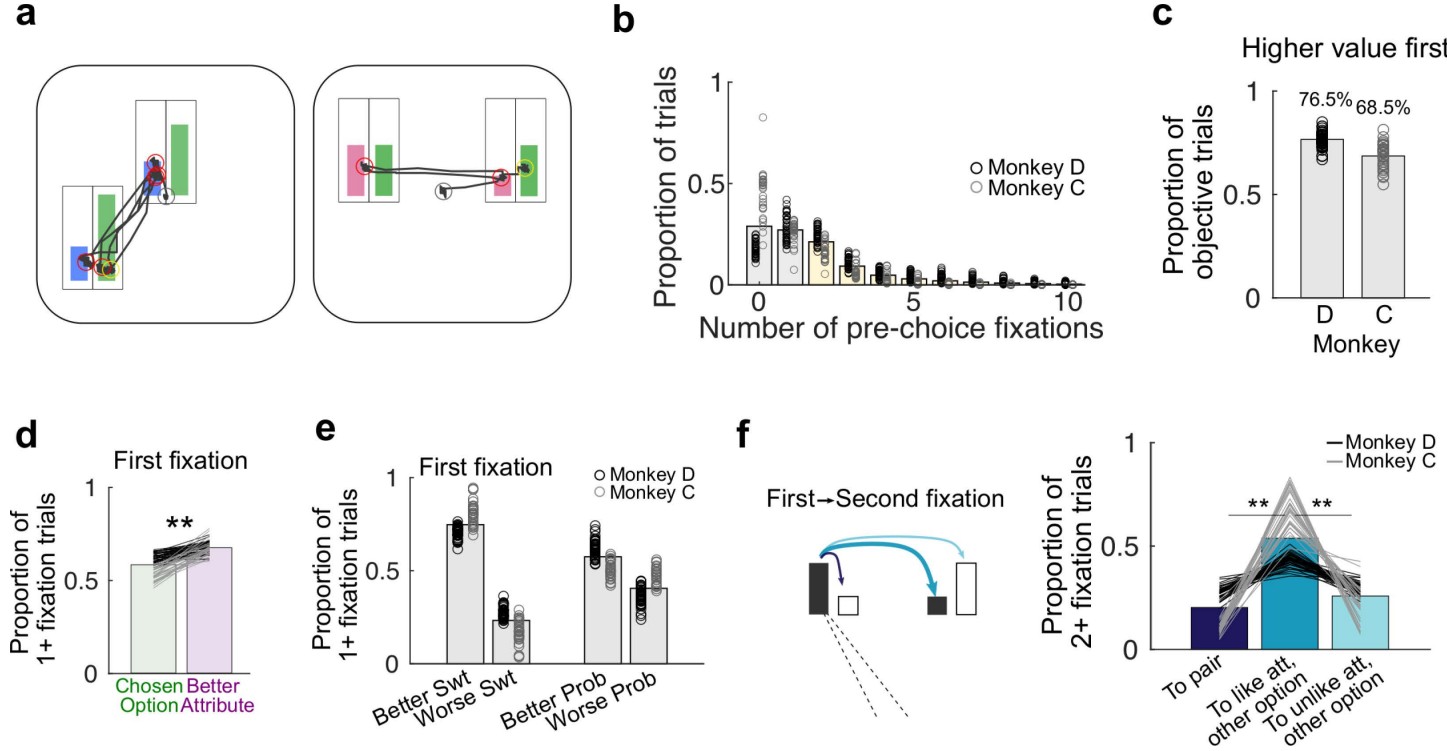

and chosen/unchosen reference frame, we found no significant relationships (Fig 2f–2g). Therefore, although encoding patterns are suggestive of a transition from encoding attributes in a pre-choice to post-choice frame of reference, this analysis did not find evidence for within-attribute comparisons in OFC.

## Gaze behavior relates to attributes and attribute values

Since our task represented attributes with separate cues, we were able to use eye tracking to assess visual exploration of attributes and whether directing gaze to an attribute affects neural responses in OFC. To do this, we defined pre-choice fixations as those directed to an attribute bar between the time of option presentation and choice (Fig 3a). Since monkeys had to fixate an option to report their choice, we excluded this final fixation that coincided with the bar release. Both monkeys visually explored the choice options, but often did not foveate all four attributes on the screen (Fig 3b). Since both

**Fig 3. Fixation patterns during multi-attribute choices. (a)** Gaze paths (black) on two example trials. Outlines around the colored bars show inclusion regions assigned to each attribute (not visible to the monkey). Red circles indicate fixations included in analyses. Fixations on the central fixation point at trial start (gray circle) and on the selected option at bar release (yellow) were excluded. **(b)** Number of pre-choice fixations made by each monkey. Bars = average across monkeys and sessions, circles = session averages. Yellow bars = trials included in the gaze-aligned neural analyses. **(c)** Proportion of trials in which both attributes of one option were higher value than both attributes of the other, and the monkey looked at the higher value option first. Bars = average across sessions, circles = average in each session. **(d)** Proportion of trials with one or more pre-choice fixation in which the monkey first looked at an attribute in the option he would ultimately choose (green) or at an attribute that was higher value than the like attribute of the other option, regardless of choice (purple). Bars = average across monkeys and sessions, lines = session averages. Black = Monkey D: Gray = Monkey C. Proportions were compared with sign-rank test across sessions. \*\*Monkeys D: $p = 1.59 \times 10^{-8}$, C: $p = 1.73 \times 10^{-6}$. **(e)** Proportion of trials in which the monkey first looked at a sweetness bar (Swt) and it was the better or worse sweetness (left), or first looked at a probability bar (Prob) and it was the better or worse probability (right). Bars = average across sessions and monkeys, circles = session proportions. **(f)** Schematic of three types of gaze transitions in an example where the first fixation is on the sweetness bar of the left option (left), and proportions of each type of gaze transition (right). Navy: to the paired attribute; Dark teal: to the like attribute of the other option; light teal: to the unlike attribute of the other option. Bar plots = average across sessions and monkeys, lines = proportions within sessions. One-way ANOVA of session-wise measures: Monkeys D: $F_{2,126} = 71.4$, $p = 1.86 \times 10^{-21}$, C: $F_{2,87} = 65.5$, $p = 4.50 \times 10^{-18}$. \*\*Bonferroni-corrected post-hoc comparisons $p < 0.001$.

attributes contributed to their choices, this suggests that subjects used both directed gaze and peripheral vision to evaluate the options. Consistent with this, the first fixation of both monkeys was more likely to be on the higher value option on objective trials, where both attributes of one option were more valuable than both attributes of the other (Fig 3c). Monkey D and C fixated the higher value option on 76.5% and 68.5% of these trials, suggesting that they rapidly evaluated peripheral targets before shifting their gaze, as monkeys do on single attribute choices [40]. However, our subjects were less proficient at this covert evaluation when attributes needed to be weighed. Across all trials, including those in which the higher value sweetness and higher value probability belong to different options, their tendencies to look first at the chosen option dropped to 62.2% and 52.7% (Monkey D, C). If the subjects evaluated unintegrated attributes, however, we might expect their first fixations to be directed to better attributes, regardless of which option was chosen. Indeed, both subjects' first fixations were directed to the better sweetness or probability on each trial more consistently than to the chosen option (Fig 3d). This tendency was stronger for sweetnesses, but also present for probability (Fig 3e). Taken together, patterns of gaze behavior indicate that subjects used peripheral vision to evaluate unintegrated attributes, and they were biased to look at valuable attributes first as they made their choices.

If the monkeys made choices by comparing like attributes, they might also sequentially evaluate the same attribute of each option. However, if their decision strategy relied on integrating the values of two attributes within an option, they might sequentially evaluate the sweetness and probability bar of a single option. To assess these possibilities, we categorized the first gaze transition on trials with at least 2 pre-choice fixations as shifting either to the paired attribute within the same option (e.g., sweetness A to probability A), to the like attribute of the other option (e.g., sweetness A to sweetness B), or to the nonmatching attribute in the other option (e.g., sweetness A to probability B) (Fig 3f). We found that gaze more frequently shifted between like attributes than between unalike attributes of either option. Therefore, patterns of gaze transitions were consistent with choice probabilities and accuracies that indicate monkeys compared unintegrated attributes.

## OFC neurons encode attributes relative to the focus of gaze

Having quantified gaze behavior, we next asked whether OFC neurons encode the value of attended attributes or options. To do this, we aligned neural activity to the onset of pre-choice fixations, and used multiple linear regressions to determine how firing rates varied with the value of attributes relative to the focus of the subjects' gaze. Each attribute of the choice was labeled as either the fixated attribute (e.g., sweetness A), the fixated attribute's pair in the same option (e.g., probability A), the attribute in the unfixated option that is like the fixated one (e.g., sweetness B), or the other attribute in the unfixated option (e.g., probability B) (Fig 4a). If OFC neurons encode attended attributes only, then we expected neuron activity to be explained by the value of the fixated attribute alone, but if they encoded the integrated value of the attended option, then firing rates should be modulated by both the fixated attribute and its pair (green attributes in Fig 4a). On the other hand, if OFC neurons compare like attributes, we expected firing rates to encode the fixated attribute and the like attribute of the unfixated option (dark green and dark orange in Fig 4a). Since evidence suggests that gaze shifts early in a decision are primarily used for evaluation and later ones are confirmatory once a latent choice is identified [41–43], we included only the first and second fixations on trials with 2 or more pre-choice fixations (Fig 3b). Further, to focus on neural activity that shifts with the targets of gaze, we first regressed out activity related to static chosen and unchosen attributes ("Materials and methods").

Soon after directing gaze to an attribute, many OFC neurons encoded attribute values relative to the focus of the monkey's gaze. In particular, neurons tended to encode the fixated attribute, often with a positive beta coefficient, and the like attribute of the unfixated option, often with a negative beta, as shown in example neurons in Fig 4b. This pattern, equivalent to a weighted subtraction, is consistent with competitive inhibition between an attended attribute and the matching attribute in the other option, and suggests that OFC neurons do carry out attribute-level comparisons. The comparison signal, however, appears to depend on directed attention, since the same signatures were not found in our previous analyses that did not take gaze into account. We considered an alternative model, which hypothesizes that neurons are selective

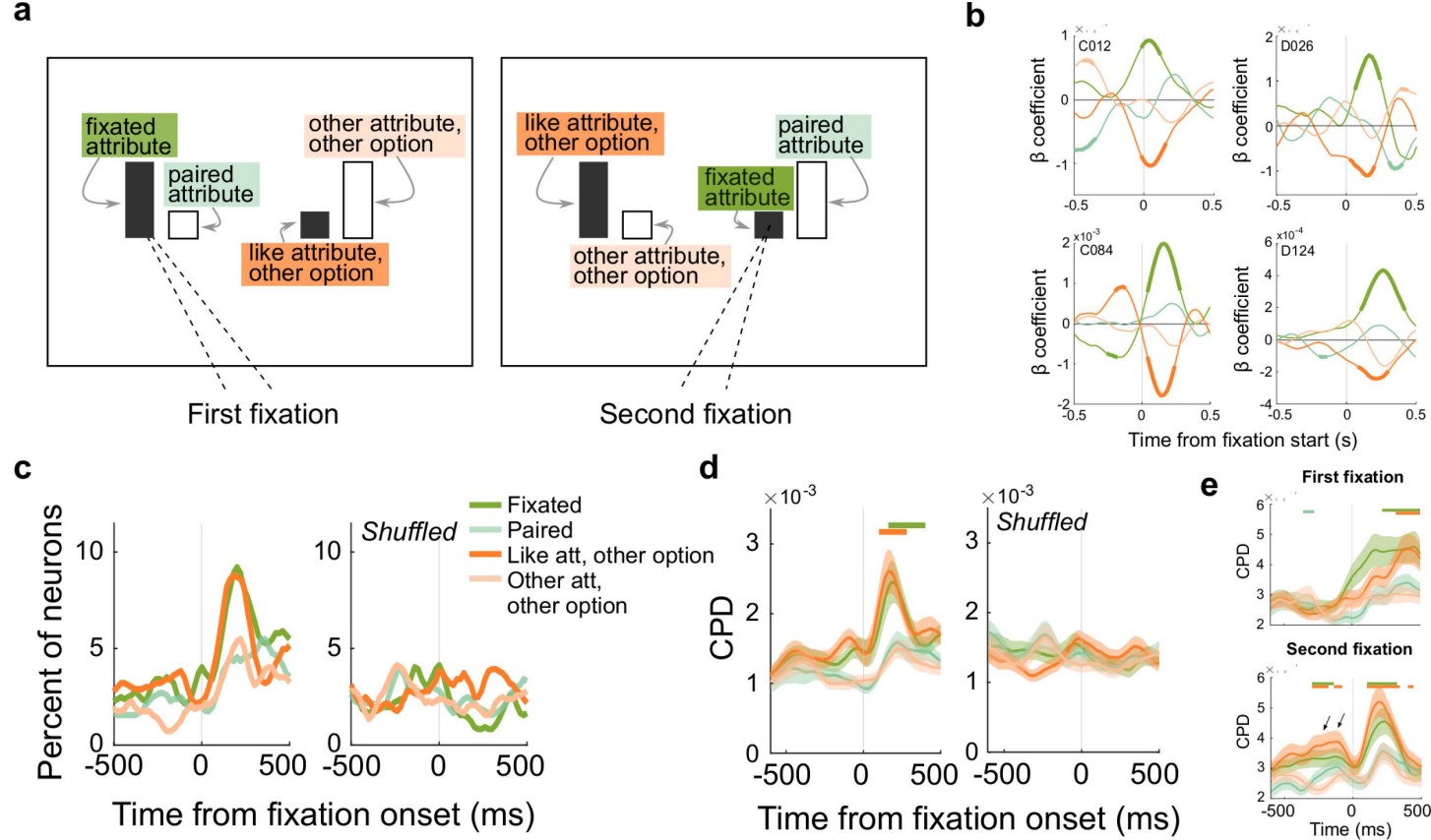

**Fig 4. OFC neurons encode attributes relative to the focus of gaze. (a)** Schematic showing attribute labels, depending on the monkeys' targets of gaze for two example fixations. The first fixation is on the sweetness bar of the left option, and the second fixation is on the sweetness bar of the right option. **(b)** Four example neurons encoding attributes in an attention-based reference frame. Plots show beta coefficients from multiple regressions, with firing rates aligned to the onset of a fixation. Thick segments denote regions of significance ($p \leq 0.05 \times 3$ consecutive time bins). Colors as in **a**. **(c)** Percent of neurons in intact (left) and shuffled data (right) that significantly encode each attribute. **(d)** CPDs for each attribute across all recorded neurons. Shading shows mean CPD ± sem. CPDs were compared to shuffled controls with rank-sum tests. Bars: $p \leq 0.05 \times 3$ consecutive time bins. **(e)** Same as **d**, except separately for first and second fixations. Arrows indicate significant encoding before the second fixation. See also S3c–S3f Fig.

for the sweetness or probability in a gazed-at option, but insensitive to which attribute of the option the fixation is directed at. However, we found that this model fit the vast majority of neurons worse than the present one, indicating that OFC neurons preferentially encode attributes relative to gaze, regardless of the identity of the attribute (S3 Fig). In the following sections, we quantify gaze-dependent attribute encoding across our recorded population of OFC neurons.

Overall, there was a clear increase in neurons encoding values of both the fixated attribute and the like attribute of the other option soon after the start of a fixation (Fig 4c). At the same time, we did not find a change in the proportion of neurons encoding the other attributes on the screen, including the paired attribute in the same option that should be encoded if the neurons integrated option values. However, the proportion of neurons that reached our threshold for significance at any time was modest (≤10%), so we also calculated the unique variance accounted for by each of the attributes across all neurons in our data set, using the coefficient of partial determination (CPD). Similar to the thresholded results, there were peaks in variance explained by the fixated attribute and the like attribute of the other option soon after the start of a fixation (Fig 4d). Therefore, at the population level, there was increased encoding of like attribute values, relative to the focus of the monkeys' gaze.

Before analyzing this effect further, we ensured that our results were related specifically to directed gaze, and were not artifacts of the analyses. First, we ran a shuffling procedure in which the same attribute values were included in each trial, but their relationship to the monkey's gaze was shuffled. In other words, we used the same residualized data, but attribute values were randomly designated the fixated attribute, paired attribute, like attribute of the other option, and the other attribute of the other option. As expected, shuffling removed the encoding that followed fixation onset in both thresholded data and CPDs (Fig 4c and 4d). Further, if only unfixated attributes were shuffled, then only encoding of the fixated attribute was found (S4g Fig).

Next, since monkeys tend to look at better attributes and chosen options first, we ensured that encoding patterns were not spurious effects of correlations between the monkeys' gaze and the value of the choice attributes. To do this, we simulated neuron responses that encoded task variables but were agnostic to the fixation patterns the monkey generated, then tested whether the analyses we used produced spurious encoding of fixated attributes. We simulated neuron responses of six types, corresponding to the most common encoding patterns found in our data set: neurons encoding chosen sweetness only, chosen probability only, better sweetness only, better probability only, chosen value (i.e., both chosen sweetness and chosen probability), and non-selective neurons (S5a Fig). We simulated firing rates that reflected only the variable being tested, plus random noise, using actual trial variables (e.g., chosen sweetness/probability on each trial) from each session, and then tested the simulated neurons on actual patterns of fixations produced by the monkey on the same trials. Using this approach, we found <1.5% of simulated neurons of each type encoded attributes relative to the monkeys' fixations (S5b–S5g Fig), consistent with the expected false alarm rate of the significance criterion we used ($p \leq 0.01$). Therefore, there was no evidence that fixation-dependent encoding arose from confounds in the behavior or analyses.

Finally, encoding of fixated and like attributes occurred approximately 200 ms after the start of a fixation. Since each fixation lasted approximately 200 ms (S4a and S4b Fig), and monkeys tended to shift their gaze between like attributes, we assessed whether encoding assigned to the like attribute of the other option was actually a consequence of the subject moving their eyes to that attribute on the second fixation. To check this, we analyzed the first and second fixations separately. Following each fixation, we obtained similar results in both proportion of neurons encoding these variables and CPDs, each occurring approximately 200 ms after the respective fixation (Figs 4e and S4c–S4f). Interestingly, on the second fixation only, there was also encoding of the two relevant attributes before the fixation was initiated. This was separate in time from post-fixation encoding, and can be attributed to the prior (first) fixation (Fig 4e). Together, these analyses confirm that OFC neurons did encode both the fixated attribute and the like attribute in the other option at the same time.

## OFC neurons jointly encode attributes

Next, we assessed whether different attributes tend to be jointly encoded by single OFC neurons, as in the example neurons in Fig 4b, or if the population effects resulted from different pools of neurons encoding a single attribute at a time. We considered four possible combinations of two attributes that might be encoded together: the fixated attribute and its pair, as expected if the neuron encoded the attended option's value (Fig 5a, pink); the fixated attribute and the like attribute of the other option, as expected if like attributes are compared (Fig 5a, blue); the two like attributes that were not fixated (Fig 5a, brown); and the two attributes of the unfixated option (Fig 5a, purple). We found a small but significant proportion of neurons jointly encoding the fixated attribute and the like attribute of the other option, consistent with the possibility of within-attribute comparisons (Figs 5b and S6b). Interestingly, we also found a small number of neurons that jointly encoded the two like attributes that are not fixated, which may be related to the monkeys' tendency to peripherally evaluate attributes. On the other hand, there was no tendency to jointly encode the two attributes of either the fixated or unfixated option, again arguing against integrated encoding of option values in this gaze-dependent reference frame. The same patterns were found when data were separated into first and second fixations or by monkey (S6c–S6d Fig).

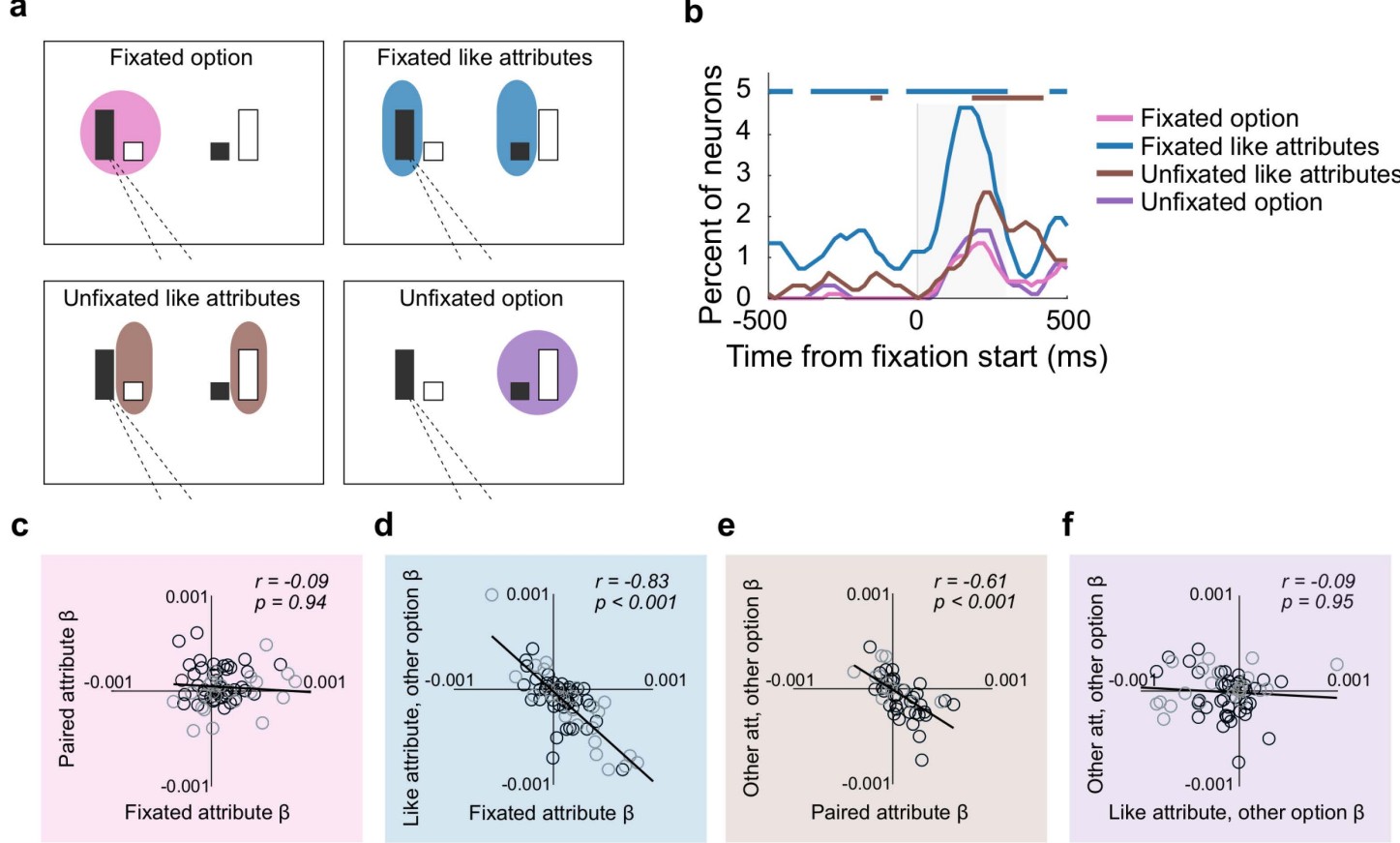

**Fig 5. Neural signatures of attribute comparison in OFC. (a)** Schematic of pairs of attributes that might be grouped in an attention-dependent reference frame. Each panel shows groupings for an example where the sweetness bar of the left option is fixated. **(b)** Percent of neurons jointly encoding each pair of attributes defined in **a**. Shuffled data are in S4b Fig. Bars = incidence of joint coding is greater than expected if the two attributes of a group were coded independently. To correct for each attribute belonging to two defined groups, significant joint encoding was defined by binomial test $p \leq 0.005 \times 3$ consecutive time bins. Shaded region = time windows analyzed in the scatter plots in **c–f**. **(c–f)** Scatter plots of mean beta coefficients of neurons with significance for one or both of the attributes in a group, color coded as in **a**. Statistics are from Pearson correlations, with *p*-values Bonferroni corrected for two comparisons each. Black circles = Monkey D, gray circles = Monkey C.

## Attribute comparison takes place in a gaze-based reference frame

Together, these results are consistent with the possibility that OFC neurons compute comparisons among like attributes during multi-attribute choice. In a final analysis, we tested for neurophysiological signatures of value comparison in this gaze-dependent reference frame. Our earlier analyses found no evidence for value comparisons, either among attributes or options (Fig 2f–2g), but this could be because OFC computes comparisons in an attention-dependent reference frame and those analyses did not take gaze into account. Therefore, we tested whether values of the unfixated attribute or option antagonistically weight value encoding of the fixated attribute/option. To do this, we computed the average beta coefficient in a time window following an attribute fixation, and included neurons that reached significance for one or both regressors of interest at any time in that window.

Consistent with comparisons of like attribute values, we found an inverse relationship between the beta coefficients of the fixated attribute and the like attribute of the other option (Fig 5d). These effects held when fixations on sweetness and probability were separated in the model (sweetness fixations: $r = -0.47$, $p = 3.8 \times 10^{-5}$, probability fixations: $r = -0.86$,

$p = 1.0 \times 10^{-14}$), and also when the comparison included all neurons without thresholding for significance (S6e–S6h Fig). In the same time window, there was no relationship between the two attributes of either the fixated or unfixated option (Fig 5c or 5f, respectively), indicating no evidence that OFC neurons encode integrated values in a gaze-dependent manner. Interestingly, there was a negative relationship among coefficients of the two like attributes that were not fixated (Fig 5e), again likely arising from the subjects' tendency to peripherally evaluate information. Since there was no relationship among attributes within the same option, the negative relationships among like attributes that were both fixated and not fixated could only arise if separate populations of neurons were modulated by each (fixated and not fixated). In addition, there was a weaker tendency for fixated attributes to be negatively related to the other attribute of the other option, raising the possibility of a small degree of comparison between the fixated attribute's value and the value of the unmatching attribute in the other option (S6i–S6j Fig). In sum, our results demonstrate a consistent neural signature of value comparisons that are primarily among like attributes and depend on the monkeys' current focus of gaze.

Overall, there was a bias among OFC neurons to encode the fixated attribute positively, meaning that firing rates increased with increasing value (Fig 5d). The like attribute of the other option, on the other hand, tended to be encoded negatively. Since the first gaze transition tended to shift between these two attributes (Fig 3f), the first fixated attribute had a high probability of subsequently becoming the like attribute of the unfixated option (Fig 6a). Therefore, we expected the beta coefficients to flip from encoding the first fixated attribute positively to negatively at the start of the second fixation, and vice versa for the like attribute of the other option. To test this, we ran a multiple regression on the firing rates aligned to the first and second fixations separately, but used regressors defined by the first fixation throughout. As predicted, approximately 200 ms after the first fixation, average coefficients across the recorded population were positive for the fixated attribute and negative for the like attribute of the unfixated option. Approximately 200 ms later, however, the signs flipped (Fig 6b). Aligning the same data to the second fixation revealed that the inverted encoding pattern peaked approximately 200 ms after the second fixation, and the same effect was apparent in each monkeys' data individually (S7 Fig). Taken together, these patterns confirm predictions of a model in which OFC neurons inversely encode like attributes of multi-attribute options, in a reference frame defined by the monkey's focus of gaze.

## Discussion

Using a multi-attribute decision-making task, we found that OFC neurons encode and compare the values of like attributes, but not integrated option values. When neural responses were aligned to trial events, neurons predominantly encoded single attributes, initially relative to each other (better versus worse), and then relative to the choice (chosen

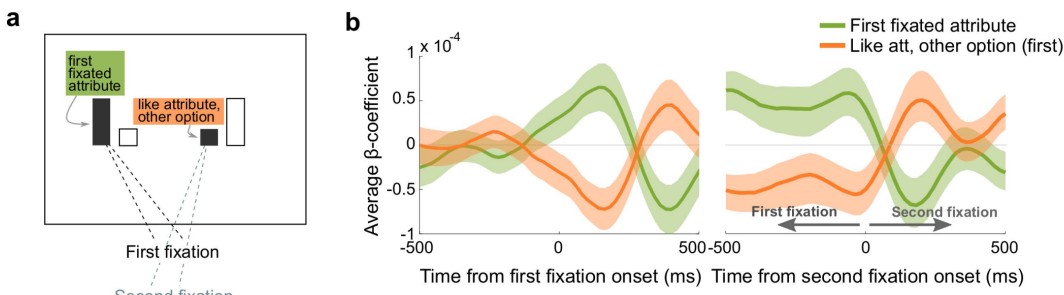

**Fig 6. Attribute encoding is modulated by gaze shifts. (a)** Schematic illustrating that a gaze transition between like attributes reverses the attribute designations. If the like attribute of the other option from the first fixation (orange) is fixated second, then it becomes the fixated attribute and the first fixated attribute (green) becomes the like attribute of the other option. **(b)** Average beta coefficients from trials with two or more pre-choice fixations. Firing rates are aligned to the first fixation (left) or the second fixation (right), with the same regression model in each. Shading shows the mean coefficient estimate across all neurons ±sem.

versus unchosen), which suggested that OFC computes choices by comparing attribute values. However, value comparisons were not anchored to the monkeys' choices nor to the attributes' relative values. Instead, the choice information to be compared was dynamically determined by the focus of the monkeys' gaze.

Our results reconcile a number of recent findings in OFC physiology, and argue against the classic view that value-based decisions are exclusively based on integrated values. Instead, they suggest a dynamic comparison process intertwined with shifts of attention, in which components of a choice option may be considered individually. The dynamic nature of this process is consistent with views of decision-making as an embodied computation, in which moment-by-moment processing is determined by an evolving interplay between one's current state and actions taken in the environment [44]. Here, as in many laboratory tasks used to study decision-making, interactions with the environment are reduced to primarily the allocation of attention. But unlike standard tasks, our design allowed us to determine when monkeys allocated attention to one attribute of a multi-attribute option. From the view of embodied decision-making, this is part of the active process of seeking information from the environment that will help determine the decision. Consistent with this idea, we found that the focus of externally-directed attention determines which choice information is encoded by OFC neurons. Moreover, these neurons encode an attended attribute value relative to that of the matching attribute in the unattended option, indicating that shifts of attention don't simply allow different choice information to be encoded, but mediate a dynamic comparison process that can contribute to decision formation.

As in previous studies, we defined value comparisons as antagonistic effects of two cue values on the firing rate of an individual neuron, measured as inverse relationships between regression coefficients for the respective cues [4,45–47]. This is similar to a value difference, which has also been used to define potential value comparisons [9,14,19,20], except the subtracted components have independent weights. This type of response indicates a competition between the cues, since firing rates are simultaneously driven up in proportion to one cue's value and down in proportion to the other. The resulting neural responses correlate with the relative value of the cues, yet the underlying circuit mechanisms that produce this signature are still unclear. One suggested mechanism is mutual inhibition among value coding neurons [4,45,48,49]. From this view, activity in a pool of neurons selectively encoding the value of one cue (in our case an attended attribute), inhibits other neurons encoding a different cue (the like attribute of the unattended option), and vice versa. Since neuron activity depends on cue value, so does their inhibitory influence, resulting in an antagonistic firing rate code. While this is possible, the gaze-dependent nature of the comparisons we found presents a challenge for the mutual inhibition model. While some OFC neurons have reliable selectivity for attended cues [22], these would have to selectively inhibit neurons encoding the unattended sweetness when the monkey fixates a sweetness bar and the unattended probability when he fixates a probability bar, and dynamically shift inhibition targets as the subject gazes around the screen. Alternatively, a more general competitive inhibition process might produce the same effects [21], as could competition arising from limited resources such as attention. For example, if the monkeys covertly divide attention between compared attributes but allocate attention in proportion to value, then higher value alternatives would subtract more attention and ultimately produce an antagonistic effect on the attended item, as described here. Relative value coding of a similar pattern can also be produced by divisive normalization, a canonical computation that normalizes one neuron's response by the activity of nearby neurons [50]. Divisive normalization of value representations has been described in the lateral intraparietal cortex for cues in versus out of a receptive field [51], as well as in medial OFC for risky and safe options [52], suggesting this is a potential mechanism in OFC as well. Interestingly, divisive normalization patterns in medial OFC were much weaker when the monkey was forced to choose one option, obviating option comparisons [52], suggesting that divisive normalization could be a mechanism underlying the comparison process. Further work is needed to test which of these candidate mechanisms computes attention-dependent comparisons in OFC.

Many studies have attempted to isolate comparison signals by presenting choice options to subjects sequentially, and therefore focusing the monkeys' attention on one option at a time [9,18,19,21,45–47]. In some regions, such as ventromedial prefrontal cortex (vmPFC) and ventral striatum, this has revealed comparison of integrated values [45,46],

but in OFC, results have been less clear [18,21]. For instance, when options have a single attribute, OFC neurons primarily encode the first or second cue value without coding the options jointly or antagonistically as expected with option comparison [19]. However, another study found value difference signals in OFC by sequentially presenting options that varied in two attributes [9]. Our results suggest that those comparison signals may actually have been among component attributes, which correlate with the overall values the authors calculated, and that sequential presentation helped to control how the monkeys' attention was allocated. Consistent with this, multi-attribute choices that do not use sequential presentation or otherwise take attention into consideration have not found strong evidence for value comparison signals [14,15,20].

The proposal that OFC neurons compare values in an attention-dependent reference frame is corroborated by a recent study that revealed attributes of a choice to the subject one at a time [22]. Antagonistic value coding was found between current and previously attended information, though this was not clearly driven by comparisons of like attributes. In the case where two like attributes were shown first, the data aligned closely with our results, but when two attributes within the same option were shown first, their values were positively correlated, consistent with an integrated value signal [22]. In contrast, our task presented all attributes simultaneously, and allowed monkeys to sample and compare information in any order. Although this results in more complicated behavior, it was able to reveal more naturalistic information sampling and decision-making strategies. In this case, we found comparison signals in OFC mainly between like attributes, and no tendency to integrate attributes within the attended or unattended option. In fact, we only found evidence for potential integration among chosen (but not unchosen) attributes later in the choice, suggesting that this signal reflects the outcome that is expected from a decision rather than the input to it. From this view, integration prior to choice, as found in previous tasks with sequential option presentations [9,22] may only occur when the other option's information is not yet available, and might reflect the current best estimate of the expected outcome.

Our results are consistent with mounting evidence that value encoding in OFC is strongly dependent on attention. This includes both overt shifts of gaze [32,35] and covert shifts of attention [33]. Moreover, attentional modulation of value encoding appears to be a more prominent feature of OFC activity compared to regions of the cingulate or lateral prefrontal cortex, where the location of attended items seems to be more relevant [22]. At the same time, we found clear evidence that monkeys also used peripheral vision to evaluate options. Their first fixations were not directed randomly, but were more often to higher value attributes, implying that they rapidly and covertly evaluated components of the display to direct their eyes [40]. It's unclear whether the attribute comparisons we report depend on peripheral perception of attributes in the unfixated option, or if this comparison is based on a stored representation of other attributes' values. One study that presented attributes sequentially, removed them from view after sampling and still found comparison signals, suggesting that OFC can make comparisons to remembered value representations [22]. However, other recent results found that removing unfixated options from view roughly doubled the size of attentional biases in human choice behavior, indicating that peripheral vision does play a large role in choice computation [53]. Whether attribute comparisons in OFC are enhanced or altered by the presence of other options in peripheral vision remains to be investigated.

Taken together, our results demonstrate that OFC is involved in evaluating and comparing component attributes of multi-attribute options. This leaves open the question of how decisions are made when options do not share common attributes. On one hand, these choices might involve different decision strategies. For instance, attributes could still be evaluated separately in an accept/reject fashion [54–56]. On the other hand, these decisions might uniquely require the use of integrated values. To date, there is stronger evidence that vmPFC, rather than OFC, is involved in comparing integrated values [45,48,52,57–59], and this process may also involve attention [57]. Going forward, conceptualizing decision formation as an embodied process that involves dynamic allocation of external attention, potentially integrated with sampling from internal memory stores [60], will help us better understand neural coding of decision variables and choice computation across regions to determine how the brain uses information to make optimal decisions.

## Materials and methods

### Subjects

Two adult male rhesus macaques (D and C) participated in the experiment. At the start of recording, they were 9 and 7 years old and weighed 10.0 kg and 8.2 kg, respectively. Each was surgically implanted a titanium headpost and cranial recording chamber made of biocompatible plastic (Monkey D – Ultem, Jerry-Rig USA. Monkey C – PEEK, Crist Instruments). The chambers were centered on stereotaxic coordinates calculated from 3T MR images obtained of each subject's brain. In Monkey C, the chamber was placed over the left hemisphere; in Monkey D, the right. All procedures were in accord with the National Institutes of Health Office of Laboratory Animal Welfare (OLAW) guidelines and were approved by the Icahn School of Medicine at Mount Sinai Animal Care and Use Committee (Animal Welfare Assurance reference number D16-00069 (A3111-01)), under protocol #2017-0084. Data and analysis code are available online [61].

**Task and behavior.** Monkeys sat in a primate chair in a darkened testing chamber, head-fixed facing an 18-inch computer monitor positioned 17 inches away. MonkeyLogic software [62,63] controlled the behavior interface. Eye position was recorded by an infrared eye tracker (ViewPoint Eyetracker USB-400, Arrington Systems) at a sampling rate of 400 Hz. This signal was oversampled by MonkeyLogic at 500 Hz.

Subjects were trained to perform a multi-attribute decision-making task. Behavior in this task has been previously reported [31], and the present data were collected from the same subjects upwards of two years later. Each trial began with the appearance of a fixation cue in the center of the screen. When the monkey gazed at the fixation point and simultaneously held a touch-sensitive bar for 550 ms, two (80% of trials) or three (20% of trials) choice options were displayed on the screen. Options were shown at 2 or 3 of 6 potential positions in a hexagonal arrangement around central fixation. Option positions were selected randomly, with the constraint that they were never in adjacent positions on the hexagon. Only 2-option trials were analyzed in the present study.

Each option was represented by a pair of two bars. The width of each bar was 2 degrees of visual angle, and the height varied from 2 to 10 degrees. The size of the left bar of each pair indicated the sweetness level (25, 50, 75, 100, or 125 mM sucrose solution) and the right bar the probability level (30, 40, 50, 60, or 70%). The colors of the bars indicated whether the size of the bar increased with increasing sweetness/probability level (direct mapping), or decreased with increasing sweetness/probability level (indirect mapping). Bar size and mapping varied randomly and independently on each trial.

While the monkey continued to hold the touch bar, he had up to 5 s to freely view the images while gaze position was tracked. They could make a selection at any time by holding gaze on any part of the desired option and releasing the touch bar. This resulted in the unselected option disappearing and triggered the reward delivery immediately after an option selection. If no option was selected within 5 s, or if the touch bar was released when gaze was not directed toward an option, this triggered a 5 s timeout, during which no reward was delivered and the screen displayed a red background. When the timeout was complete, the next trial could be initiated. Incomplete trials were excluded from analysis. Inter-trial intervals were 1 s.

Rewards consisted of 0.33 mL of fluid delivered over 500 ms for Monkey D, and 0.297 mL delivered over 450 ms for Monkey C. Amounts were titrated during pretraining to account for each subject's relative weighting of sweetness and probability, and balance as closely as possible their subjective weighting of the two attributes.

### Behavior models

Choices in each session were fit with two potential models that varied only in how attribute values were combined. In the additive model (Eq. 1), sweetness and probability were independently parameterized and linearly combined.

$$
\begin{aligned}
X = \beta_0 + \ &\beta_{ProbA}(ProbA) + \ \beta_{SwtA}(SwtA) + \beta_{ProbB}(ProbB) + \ \beta_{SwtB}(SwtB) \\
&+ \ \beta_{SwtDirA}SwtDirA + \ \beta_{SwtDirB}SwtDirB + \beta_{ProbDirA}ProbDirA + \ \beta_{ProbDirB}ProbDirB \\
&+ \ \beta_{XPosA}XPosA + \beta_{YPosA}YPosA + \ +\beta_{XPosB}XPosB + \beta_{YPosB}YPosB
\end{aligned}
\tag{1}
$$

In the multiplicative model (Eq 2), attributes were independently parameterized and multiplicatively combined.

$$X = \beta_0 + \beta_{Prob}\left(\log\frac{ProbA}{ProbB}\right) + \beta_{Swt}\left(\log\frac{SwtA}{SwtB}\right) + \beta_{SwtDirA}SwtDirA$$
$$+ \beta_{SwtDirB}SwtDirB + \beta_{ProbDirA}ProbDirA + \beta_{ProbDirB}ProbDirB$$
$$+ \beta_{XPosA}XPosA + \beta_{YPosA}YPosA + + \beta_{XPosB}XPosB + \beta_{YPosB}YPosB \tag{2}$$

In both models, options were arbitrarily designated A and B. *ProbA or B* and *SwtA or B* are the ordinal magnitudes of probability and sweetness available in each option. $\beta_{Prob/Swt}$ are fitted weights of probability/sweetness attributes. *(Prob/Swt)Dir(A/B)* is the mapping for each attribute bar, coded as −1 or 1, with the corresponding coefficient ($\beta_{(Prob/Swt)Dir(A/B)}$). *XPos(A/B)* is the *x*-position of option A or B on the task screen, and *YPos(A/B)* is its *y*-position. Since attributes within options were always proximal to each other and therefore correlated, only *x* and *y* coordinates of the center point of the overall option was used. $\beta_{XPos(A/B)}$ and $\beta_{YPos(A/B)}$ are the corresponding coefficients. $\beta_0$ is a constant to capture any bias toward A or B.

For both models, the probability of choosing an arbitrary option A was fit by generalized linear models with the logit link function. This is equivalent to a softmax decision rule [64] that probabilistically selects the option with the higher value (Eq. 3).

$$Pr(Choice\ A) = \frac{1}{(1 + e^{-X})} \tag{3}$$

## Behavior model recovery

We simulated 100 sessions of 1,000 choice trials each. As in choices presented to the monkeys, sweetness and probability magnitudes and mappings were randomly selected on each simulated trial, and each simulated option was assigned an X, Y position following the same constraints in the main task. Then, for each session, we used the parameters from the GLMs fit to the monkeys' actual choices to predict a probability of selecting option A (Matlab function *predict.m*). This was done separately for additive and multiplicative models to obtain two vectors of trial-wise predictions, one for each model. For each set of predicted probabilities, we simulated choice outcomes on each trial by selecting option A with the probability *p*, and option B with probability 1 − *p*, where *p* is the probability predicted by the model on that trial. Each simulated session was then fit by the same additive and multiplicative GLMs as the original behavior to determine whether the model comparison would reliably select the appropriate additive and multiplicative model when the generative model was known.

## Objective choice accuracies and condition-wise choice models

Among trials in which both attributes of one option were superior to both attributes of the other, accuracies were calculated as the proportion of trials per session in which the sweeter and more probable reward was selected, and one-way ANOVAs compared the average accuracies per session across the subsets of trials (as in Fig 1d). To model all choices in a condition (not just those with an objectively better option), trials with different mappings within options or within attributes were aggregated across all sessions. From each condition, we created 100 bootstrapped samples of 400 trials and fit each set of trials with a simplified version of the additive model (Eq 4).

$$X = \beta_0 + \beta_{ProbA}(ProbA) + \beta_{SwtA}(SwtA) + \beta_{ProbB}(ProbB) + \beta_{SwtB}(SwtB) \tag{4}$$

Beta coefficients for the same attribute of option A and B were approximately equal and oppositely signed, so we took the average absolute value as the estimated weight for each attribute. Weights were compared across mapping conditions with Wilcoxon rank-sum tests.

## Neural recording

Neurophysiological methods were similar to those reported previously [65]. Data were collected with tungsten micro-electrodes (FHC) and 16-contact linear arrays (Plexon) lowered through chronic craniotomies, and advanced manually using custom-built microdrives. Between 4 and 14 electrodes or 1–4 16-channel probes were lowered per session. Target regions were identified on previously obtained 3T MR images, as the cortex between medial and lateral orbital sulci. Our recording areas range from 35/42.7 mm to 28/32.7 mm AP for Monkeys D/C, relative to the interaural line. This included anterior regions putatively identified as Area 11 and posterior regions putatively identified as Area 13. After recording, electrode placements were reconstructed from MRIs, and any neurons recorded outside the target regions were removed from further analysis.

Neural signals were collected, digitized, and saved (Ripple Neural Systems Grapevine Processer). Putative spikes were captured at 30 K and sorted into single and multi-units offline (Plexon OfflineSorter) and spike times were saved at 1 kHz resolution. Only single units were analyzed in the present study, and any unit with an overall firing rate <1 Hz was excluded from further analysis due to difficulty statistically characterizing their responses.

## Trial-aligned encoding models and model comparison

Prior to analysis, spike times were smoothed with a 150 ms boxcar. Smoothed firing was then aligned to the appearance of choice options and analyzed in 200 ms windows, stepped forward by 20 ms. Full time windows analyzed included 1 s before until 1s after option onset, resulting in 90 overlapping time bins. In each time bin, (Eq 5) was fit to the average firing rate. ProbCh and SwtCh are the ordinal magnitudes of the chosen probability and sweetness. ProbUCh and SwtUch are the ordinal magnitudes of the unchosen probability and sweetness. ProbDirCh and SwtDirCh are the mapping (direct or indirect) of the probability and sweetness attribute of the chosen option, and ProbDirUch and SwtDirUch are the mappings of the unchosen attributes. Each regression term was centered on zero before fitting the model.

$$
\begin{aligned}
FR = {} & \beta_0 + \beta_{ProbCh}(ProbCh) + \beta_{SwtCh}(SwtCh) + \beta_{ProbUCh}(ProbUch) + \beta_{SwtUch}(SwtUch) \\
& + \beta_{ProbDirCh}(ProbDirCh) + \beta_{SwtDirCh}(SwtDirCh) + \beta_{ProbDirUCh}(ProbDirUch) \\
& + \beta_{SwtDirUch}(SwtDirUch)
\end{aligned}
\tag{5}
$$

Significant encoding of a model term was defined as a non-zero beta coefficient. To compare encoding patterns consisting of conjunctions of attributes (e.g., both chosen attributes) to single attribute encoding, we defined significance thresholds based on a total false discovery rate of $p \leq 0.01$ in a given time bin, then required significance for at least 3 consecutive time bins. We selected this criterion because it produced false discovery rates during the fixation epoch of <5%. Single attribute encoding included any one of four model terms reaching threshold, so the significance criterion was $p \leq 0.0025$ for 3 time bins. For chosen option encoding, chosen sweetness and chosen probability must both be significant (likewise for unchosen option encoding), meaning the effective alpha is $alpha_{sweetness} * alpha_{probability}$. To achieve an effective alpha of 0.01, the threshold for each attribute was $p \leq 0.1$ for three time bins. For like attribute encoding (*i.e.*, both sweetnesses *or* both probabilities), the alphas of two conjunctive patterns are added, such that the effective alpha is $(alpha_{sweetnessA} * alpha_{sweetnessB}) + (alpha_{probabilityA} * alpha_{probabilityB})$. To achieve an effective alpha of 0.01, the significance criterion for each term was $p \leq 0.0705$ for three time bins. Equating the significance criteria for each model term, rather than the effective alphas, produced qualitatively similar results, except there were unequal false discovery rates prior to options onset.

The same neuron firing rates in the same sliding windows were also fit with additional models. In each model, regressors were mean-centered before fitting. In the Better/Worse model (Eq. 6), ProbBet and SwtBet are the ordinal magnitudes of the higher value (better) probability and sweetness, respectively and ProbWrs/SwtWrs are the ordinal magnitudes of the lower value (worse) probability and sweetness, as in Fig 2b. If an attribute had equal values in each option, then the

same number was entered into each regressor. ProbDirBet and SwtDirBet are the mapping (direct or indirect) of the better probability and sweetness, and ProbDirWrs and SwtDirWrs are the mappings of the worse probability and sweetness.

$$FR = \beta_0 + \beta_{ProbBet}(ProbBet) + \beta_{SwtBet}(SwtBet) + \beta_{ProbWrs}(ProbWrs) + \beta_{SwtWrs}(SwtWrs) + \beta_{ProbDirBet}(ProbDirBet)$$
$$+ \beta_{SwtDirBet}(SwtDirBet) + \beta_{ProbDirWrs}(ProbDirWrs) + \beta_{SwtDirWrs}(SwtDirWrs)$$

(6)

In the additive chosen value model (Eq. 7), attributes were first weighted by their relative importance in the additive behavior model for that session (Eq. 1). Since options A and B were arbitrarily designated in the behavior model, their weights were approximately equal and opposite. Therefore, we took the mean absolute value of the behavior coefficients for probability attributes ($\beta_{BehProb}$) and multiplied this by trial-wise probabilities, the mean absolute value of behavior coefficients for sweetness attributes ($\beta_{BehSwt}$) multiplied by trial-wise sweetnesses, then added these to obtain a weighted additive value for chosen and unchosen options. The model also included attribute mappings as in (Eq. 5).

$$FR = \beta_0 + \beta_{Ch}(\beta_{BehProb}ProbCh + \beta_{BehSwt}SwtCh) + \beta_{Uch}(\beta_{BehProb}ProbUch + \beta_{BehSwt}SwtUch)$$
$$+ \beta_{ProbDirCh}(ProbDirCh) + \beta_{SwtDirCh}(SwtDirCh) + \beta_{ProbDirUch}(ProbDirUch)$$
$$+ \beta_{SwtDirUch}(SwtDirUch)$$

(7)

In the multiplicative chosen value model (Eq. 8), attributes were weighted by their relative importance in the multiplicative behavior model for that session (Eq. 2). This behavior model estimated one coefficient for the log ratio of probabilities ($\beta_{Prob}$), and another for the log ratio of sweetnesses ($\beta_{Swt}$). These were used as weights for each attribute, and attributes of chosen and unchosen options were multiplied to obtain integrated values. The model also included attribute mappings as in (Eq. 5).

$$FR = \beta_0 + \beta_{Ch}(\beta_{Prob}ProbCh * \beta_{Swt}SwtCh) + \beta_{Uch}(\beta_{Prob}ProbUch * \beta_{Swt}SwtUch)$$
$$+ \beta_{ProbDirCh}(ProbDirCh) + \beta_{SwtDirCh}(SwtDirCh) + \beta_{ProbDirUch}(ProbDirUch)$$
$$+ \beta_{SwtDirUch}(SwtDirUch)$$

(8)

Finally, we considered the possibility that neurons encoded risk, or uncertainty, rather than probability [66]. The risk model (Eq. 9) was identical to (Eq. 5), except the ordinal levels of probability were replaced with ordinal levels of uncertainty, such that probabilities 1, 2, 3, 4, 5 were replaced with 1, 2, 3, 2, 1. RiskCh and RiskUch are chosen and unchosen risk attributes, respectively.

$$FR = \beta_0 + \beta_{RiskCh}(RiskCh) + \beta_{SwtCh}(SwtCh) + \beta_{RiskUCh}(RiskUch) + \beta_{SwtUch}(SwtUch)$$
$$+ \beta_{ProbDirCh}(ProbDirCh) + \beta_{SwtDirCh}(SwtDirCh) + \beta_{ProbDirUCh}(ProbDirUch)$$
$$+ \beta_{SwtDirUch}(SwtDirUch)$$

(9)

Each of the encoding models above were used to explain trial-wise variance in firing rates in 200 ms sliding windows. In each window, the total model significance was computed, as well as the Akaike information criterion (AIC). If the same model had a significance of $p \leq 0.01$ for three consecutive time bins and had the lowest AIC of the models under consideration (Eqs 5–7) in the main text, (Eqs 5–9) in S2i Fig), then it was included as a neuron best fit by that model in the relevant time bins. Note that this means that the same neuron could be assigned to different models in different time bins. To determine whether single neurons initially coding better/worse attributes were more likely to code chosen/unchosen attributes later in the trial, we found the proportion of neurons assigned to the better/worse model in any time bin in the first 350 ms after the choice appeared, and the proportion assigned to the chosen/unchosen model in any time bin between 350 and 700 ms and tested whether the probability of a neuron significantly encoding both of these was higher than

expected if the two models were encoded independently (i.e., the joint probability of the two types of encoding). The rate of actual joint encoding was compared to this chance level with binomial tests. To investigate relationships between beta coefficients for different encoded attributes, we averaged coefficients from the sliding regressions in time windows with a leading edge 100–400 ms after the appearance of the choice options, and counted neurons with significance in any of the included bins as encoding neurons. This time range was chosen to encompass the majority of pre-choice data (the faster monkey had a median choice time of 520 ms), while allowing for visual processing delays after the options appeared and motor processing times before a choice was registered.

## Gaze analyses

The EyeMMV toolbox [67] was used to define fixations from continuous eye movements. The minimum fixation duration was 50 ms to capture short fixations [68]. The tolerance to include a gaze position in a fixation cluster ($t1$) and to include it in the calculation of the fixation cluster mean ($t2$) were $t1 = 2$ and $t2 = 1$. Only fixations that fell within defined regions around each attribute bar were included in the analyses. To capture fixations that fell on the bar edges, these regions included the full bar height plus 1.0° of visual angle around all sides except the side adjacent to the other attribute, which was 1.0° away. Here, an additional 0.5° buffer was used, so there was no unassigned space between the bars. The eye tracker was calibrated at the start of each session, and gaze data were further aligned in post-processing by centering eye traces on the mean *x-y* coordinates during the initial fixation window on each trial. Analyses excluded the fixation that coincided with release of the touch bar (i.e., the choice report). For each pre-choice fixation, EyeMMV was used to quantify fixation position (in *x*, *y* coordinates), start time, end time, and duration. Fixation start times were used to align neural data to fixations. Except where noted, the first and second fixations of trials with 2 or more pre-choice fixations were used.

Neural responses aligned to fixations and analyzed similarly to trial-aligned data. Firing rates were smoothed with 150 ms boxcar, averaged in sliding windows of 200 ms stepped forward by 20 ms, and analyzed with GLMs. To remove fixation-unaligned effects, we first fit each time window for each neuron with (Eq 5), then fit the residuals with (Eq 10), where FixVal and FixDir are the ordinal value and mapping direction of the fixated attribute (e.g., sweetness A), PairVal and PairDir are the ordinal value and mapping of the attribute that is the other component of the fixated option (e.g., probability A), LikeAttOtherOptVal and LikeAttOtherOptDir are the value and mapping of the like attribute of the unfixated option (e.g., sweetness B), and OtherAttOtherOptVal and OtherAttOtherOptDir are the value and mapping of the other attribute of the unfixated option (e.g., probability B). FixAtt indicates whether the currently fixated attribute is sweetness or probability, and XPosFix and YPosFix are the X and Y position coordinates of the fixation. Significant encoding was defined as $p \leq 0.05$ for three consecutive time bins. This criterion was selected because it resulted in <5% false discovery rates in shuffled data. Results were qualitatively similar if a stricter criterion was used ($p \leq 0.01$ for three consecutive time bins).

$$
\begin{aligned}
FRresid = {} & \beta_0 + \beta_{FixVal}(FixVal) + \beta_{PairVal}(PairVal) + \beta_{LikeAttOtherOptVal}(LikeAttOtherOptVal) \\
& + \beta_{OtherAttOtherOptVal}(OtherAttOtherOptVal) + \beta_{FixDir}(FixDir) + \beta_{PairDir}(PairDir) \\
& + \beta_{LikeAttOtherOptDir}(LikeAttOtherOptDir) + \beta_{OtherAttOtherOptDir}(OtherAttOtherOptDir) \\
& + \beta_{FixAtt}(FixAtt) + \beta_{XPosFix}(XPosFix) + \beta_{YPosFix}(YPosFix)
\end{aligned}
\tag{10}
$$

CPD quantifies the increase in unexplained variance obtained by removing one predictor from a multiple regression model, and therefore measures the unique variance explained by that predictor. CPDs were calculated for all neurons in the same sliding windows as above.

Joint encoding was defined as significant encoding (also $p \leq 0.05$ for three consecutive time bins) of two regressors in the same time bin. If the two regressors were encoded independently in a population, more coincident encoding would occur by chance as the prevalence of encoding either increased. This rate of chance joint encoding was calculated as the

product of the proportions of neurons encoding either regressor, and the actual rate of coincident encoding was compared to chance with binomial tests. To quantify the tendency to code two regressors similarly or inversely, we averaged the regression coefficients for pairs of regressors from (Eq 10) in time windows where the leading edge was >0 ms and <300 ms from the start of fixation. We included any neuron that reached significance for encoding either of the regressors in one or more time windows in that range. Note that fixated and non-fixated attributes of both types had the same ordinal range in this analysis (1–5, mean centered to −2 to +2), so unequal range effects cannot explain beta anticorrelations in this task [21].

To test whether our main effects held when fixations on sweetness and probability were separated, we tested a model that removed the identity regressor and separated fixations on sweetness and probability into two predictors (Eq. 11). In this model, FixSwt and FixProb are the value of the fixated sweetness or probability, respectively, with 0 entered when a fixation was not directed at the relevant attribute.

$$
\begin{aligned}
FRresid = {} & \beta_0 + \beta_{FixSwt}(FixSwt) + \beta_{FixProb}(FixProb) + \beta_{PairVal}(PairVal) \\
& + \beta_{LikeAttOtherOptVal}(LikeAttOtherOptVal) + \beta_{OtherAttOtherOptVal}(OtherAttOtherOptVal) \\
& + \beta_{FixDir}(FixDir) + \beta_{PairDir}(PairDir) + \beta_{LikeAttOtherOptDir}(LikeAttOtherOptDir) \\
& + \beta_{OtherAttOtherOptDir}(OtherAttOtherOptDir) + \beta_{FixAtt}(FixAtt) + \beta_{XPosFix}(XPosFix) \\
& + \beta_{YPosFix}(YPosFix)
\end{aligned}
\tag{11}
$$

To test the alternative possibility that single neurons were selective for the value of sweetness or probability in the option that was fixated regardless of which attribute in that option the monkey directed its eyes to, we fit neurons to a second model (Eq. 12). In this model, FixOptSwt and FixOptProb are the magnitude of the sweetness and probability bar, respectively in the fixated option; UnfixOptSwt and UnfixOptProb are the sweetness and probability magnitudes in the unfixated option; FixOptSwtDir, FixOptProbDir, UnfixOptSwtDir, and UnfixOptProbDir are the mappings of each of these attributes using the same convention. The remaining regressors are as in (Eq. 10).

$$
\begin{aligned}
FRresid = {} & \beta_0 + \beta_{FixOptSwt}(FixOptSwt) + \beta_{FixOptProb}(FixOptProb) + \beta_{UnfixOptSwt}(UnfixOptSwt) \\
& + \beta_{UnfixOptProb}(UnfixOptProb) + \beta_{FixOptSwtDir}(FixOptSwtDir) \\
& + \beta_{FixOptProbDir}(FixOptProbDir) + \beta_{UnfixOptSwtDir}(UnfixOptSwtDir) \\
& + \beta_{UnfixOptProbDir}(UnfixOptProbDir) + \beta_{FixAtt}(FixAtt) + \beta_{XPosFix}(XPosFix) \\
& + \beta_{YPosFix}(YPosFix)
\end{aligned}
\tag{12}
$$

Since (Eqs 10 and 12) have the same number of parameters, we compared them by computing the whole model $R^2$ for each model fit to each neuron, averaging the sliding analysis in the same time window as shown in Fig 5c–5f. Since nearly all neurons had higher $R^2$ for (Eq 10), we did not consider (Eq. 12) further.

### Neuron simulations

Simulations were used to tested whether neurons that varied only with trial-wise variables could produce spurious encoding of fixated attributes, given the behavior patterns of our monkeys in this task. For simulated neurons, we used actual trial variables from each behavioral session, including sweetness and probability of the choice options, the monkey's selection, and the first fixated attribute. We then simulated 5 neurons of each type (see Fig 5) per session, resulting in 215 and 150 simulated neurons of each type for Monkey D and C, respectively. Each neuron was simulated as a single vector of trial-wise firing rates that was seeded with an integer response (1–5) that depended on the task variable being tested. Non-selective neurons were randomly assigned an integer 1–5. Chosen value responses were determined by binning the additive combination of chosen sweetness and probability into 5 bins. The integer indicated which of 5 partially

overlapping, uniformly distributed ranges a new seed, representing an average firing rate, was drawn from. These ranges were chosen to match typical firing rates of OFC neurons, with the lowest value set to 0 and the highest 6.5. The average value of each range stepped up from 1 to 5, such that random draws from each range would produce average responses that were ordered $1 < 2 < 3 < 4 < 5$. These responses were inverted (to $5 < 4 < 3 < 2 < 1$) with a probability of 0.5 to simulate the tendency of OFC neurons to encode value both positively and negatively. We then added two sources of gaussian noise to each trial: one that was unique to each simulated neuron and one, on average 10% as large, that was shared across simulated neurons in the same session. Depending on the simulated variable, (Eq. 5) or (Eq. 6) was used to confirm that the simulated neurons had the desired encoding properties. We then tested whether the same analyses used to show fixation-related encoding (described above) would produce spurious results.

## Supporting information

**S1 Fig. Behavior is best fit by models with independent attributes. (a)** AIC (left) and BIC (right) comparing behavior models in which attributes were combined additively or multiplicatively. Each point is a session, diagonal line = unity. In nearly every session, the additive model resulted in lower AIC and BIC values, indicating a better fit. **(b)** A model recovery procedure (see "Materials and methods") ensured that additive and multiplicative models produced choices that could be reliably differentiated. Data were simulated from either a multiplicative or additive model, then simulated data were fit with each model. AIC was used to compare fits to determine whether our analyses could reliably distinguish the correct generative model. For each animal (Monkey D left, Monkey C right), AICs were lower when the fitting model matched the generating model in every session. Bars show across-session means, lines show individual sessions. **(c–f)** GLM coefficients from additive models fit to choice behavior for Monkey D. **(c)** Significance of each coefficient estimate, shown as − log of the *p*-value calculated from the *t*-statistic testing for a non-zero coefficient. Bars show across-session means, circles show values from individual sessions. Coefficients labeled as in (Eq. 1), where A and B refer to option A and B, respectively, assigned arbitrarily in the GLM that predicted the probability of choosing option A. Attribute values had the largest effects on choices, and attribute mappings (labeled Dir) had almost no effects. **(d)** Histograms of coefficient estimates across sessions for the magnitude of each attribute. Positive/negative coefficients on option A/B indicate that the probability of choosing option A increases/decreases with the magnitude of that attribute. Larger deviations from 0 are indicative of more influence on the choice. **(e)** Histograms of coefficient estimates across sessions for the mapping of each attribute, as in **c**. **(f)** Histograms of coefficient estimates across sessions for the X and Y location of each option. Since attributes were always paired spatially, a single center location was used for each option. **(g–j)** Same as **c–f**, for Monkey C. (EPS)

**S2 Fig. Heterogeneity of single unit encoding.** If single neurons in OFC construct option values, we expect to see neurons that encode both attributes of an option in a similar manner (i.e., with the same sign), and in approximately the proportion that each attribute is weighed in patterns of behavioral choices. **(a–d)** Examples of how many recorded neurons did not conform to these expectations. Thick lines = significant encoding ($p \leq 0.01 \times 3$ consecutive time bins) in a sliding multiple regression model predicting neuron firing rates from chosen and unchosen attributes (Eq 5). **(a)** Neurons encoding both attributes of the chosen option, but in manners inconsistent with this definition of an integrated value signal. The first two neurons encode the value of both attributes nearly identically, although behaviorally both subjects weighted probability slightly more than sweetness. These neurons may instead encode the ordinal rank of the chosen attributes. The second two neurons encode both attributes but with different time courses, also inconsistent with integration. **(b)** Further examples of neurons encoding only one attribute of the chosen option. **(c)** Further examples of neurons encoding both sweetnesses (chosen and unchosen) or both probabilities. **(d)** Examples of neurons with complex encoding patterns involving multiple chosen and unchosen attributes. **(e)** Tallies of neurons that exhibited different encoding patterns in the regression model that included chosen and unchosen attributes (Eq. 5), separately for each subject. For conjunctive

encoding, significance thresholds were adjusted to equate chance levels (see "Materials and methods"). The greatest proportion of neurons in both subjects encoded a single attribute at a time (green), and both subjects showed a small peak of neurons encoding two like attributes soon after the options appeared (purple). A small proportion of neurons encoding both attributes of the chosen option appeared slightly later (yellow). In contrast, there were almost no neurons encoding both attributes of the unchosen option, suggesting a lack of unchosen integrated value signals (brown). There were also very few neurons encoding two attributes with opposite signs (orange), as expected if the comparisons between attributes were encoded. **(f–h)** Beta coefficients from the sliding regressions in **(e)**, averaged across time windows with a leading edge 100–400 ms following the appearance of choice options. Neurons significantly modulated by chosen sweetness **(f)**, chosen probability **(g)** or both **(h)** in any of the included time bins are shown separately. Neurons considered selective for single attributes by these criteria (greens) had no overall tendency for firing rates to be modulated by the other attribute. **(i)** Proportion of neurons in each subject best fit by different encoding models across time. For a neuron to be assigned to a model, it had to both predict significant variance in the neuron's activity (full model $p \leq 0.01 \times 3$ consecutive time bins) and provide the best fit as assessed by AIC. Here, we tested two additional models to the figure in the main text. In the first (red), chosen and unchosen probability was replaced with chosen and unchosen risk (or uncertainty), which was highest at 50% probability and lowest at both the lowest and highest probability. The second (blue) was a variant of the integrated value model, in which option values were computed as the multiplicative combination of weighted sweetness and probability, where the weights were determined by the attribute weights in the multiplicative behavior model for that session. Overall, the most commonly selected models were better/worse attribute values early in the choice, and chosen/unchosen attribute values later.
(EPS)

**S3 Fig. OFC encodes attended attributes, not attributes of an attended option.** We tested the possibility that OFC neurons selectively encode sweetness or probability, modulated by attention to an option. **(a)** Schematics of the two models that were contrasted, showing two fixation events. The model on the top assumes that neurons encode the value of any attribute in the focus of gaze and all other attributes on the screen are encoded in relation to that attribute. The model on the bottom assumes that neurons consistently encode either the sweetness or probability of the option that is fixated, regardless of which attribute of that option gaze falls on. **(b)** Scatterplots of whole model $R^2$'s show that 84.7% (Monkey D) and 82.4% (Monkey C) of neurons are better fit by the Fixated Attributes Model. Red circles = neurons that met criteria for encoding either the fixated attribute or the like attribute of the other option in the main paper. Gray = all other neurons.
(EPS)

**S4 Fig. OFC encoding of attributes in an attention-based reference frame. (a)** Histograms of the time that the first (blue) and second (red) fixation begins, relative to the time that the choice options are presented, plotted separately by monkey. **(b)** Histograms of the duration of the first (blue) and second (red) fixations, plotted separately by monkey. **(c)** Percent of neurons that significantly encode each attribute during only the first fixation of trials with 2 or more pre-choice fixations (i.e., the same trials analyzed in the main results). Left panel shows encoding in intact data, right panel in the same trials with fixated attribute labels shuffled. **(d)** CPDs for shuffled data shown in Fig 4e. **(e–f)** same as **c–d**, except for the only the second fixation of the same trials. Note that there are two epochs of significant encoding, one occurring after the second fixation begins, and one occurring before the start of the second fixation (arrows, also Fig 4e). The latter overlaps in time with the first fixation on these trials. **(g)** Plots as in **c–d**, of the same data as shown in the main results, except the values of the 3 unfixated attributes were randomly assigned to Paired, Like attribute other option, or Other attribute other option. Since only the fixated attribute assignment was intact, this was the only encoding recovered above chance.
(EPS)

**S5 Fig. Simulated neurons encoding trial-wise attributes do not encode fixated attributes. (a)** Two examples each of six types of simulated neurons. Average firing rates (±sem) of the simulated neurons are plotted at five values of each

attribute. Plots show firing rates varying with chosen attributes or better attributes, depending on which type of encoding was simulated. For the first five types, the top row shows simulated neurons that positively encode the target attribute(s), and the bottom row shows negatively encoding simulated neurons. Nonselective neurons are neither positive nor negative. **(b)** The population of simulated neurons encoding chosen sweetness. The first plot shows the percent of neurons encoding chosen sweetness (Ch Swt) or chosen probability (Ch Prob) on multiple linear regression as in the main data set. Significance = non-zero coefficients in the regression ($p \leq 0.01$). The second plot shows that beta coefficients for chosen sweetness and chosen probability produced the expected encoding pattern. The third plot shows the percent of neurons encoding fixation-related variables, after the simulated firing rates were residualized for trial-wise variables as in the main data set. Bars in the first and third panels are percents across both monkeys, black circles = Monkey D, gray circles = Monkey C. **(c–g)** The same as **b**, except for simulated encoding of chosen probability **(c)**, better sweetness **(d)**, better probability **(e)**, chosen value **(f)**, or no task-related variables **(g)**.
(EPS)

**S6 Fig. OFC neurons jointly and inversely encode attended attributes. (a)** The same schematic as in the main paper, showing pairs of attributes that were assessed for joint encoding, reproduced here for reference. **(b)** Joint encoding in the same data as in the main results, except with fixation assignments shuffled. **(c)** Joint encoding of attributes computed separately on the first (left) or second (right) fixation of trials with 2 or more pre-choice fixations. Two-sided binomial tests were performed on each sliding window to compare the incidence of joint coding to chance, which was defined as the joint probability of encoding either attribute of a pair. Bars indicate time windows that reached significance ($p \leq 0.005 \times 3$ consecutive time bins). Note that there was significant joint encoding both after and before to the second fixation (but not the first), the latter overlaps with the time of the first fixation. **(d)** Joint encoding of attributes computed separately by monkey. The most common joint encoding in both animals were the fixated attribute and the like attribute of the other option (blue). **(e–h)** Scatterplots of the comparisons shown in [Fig 5c–5f], including all neurons regardless of significance. Black = Monkey D, Gray = Monkey C. Pearson r and Spearman rho statistics are shown. *P*-values, Bonferroni corrected for 3 comparison each, are **(e)** both $p > 0.3$ **(f)** both $p < 10^{-10}$ **(g)** both $p < 10^{-10}$ **(h)** $p > 0.9$. **(i)** Schematic and correlation for the comparison between the value of the fixated attribute and the non-matching attribute of the other option. Scatterplot includes only neurons significant for one or both of the compared attributes, as in the main paper. **(j)** The same scatterplot as **(j)**, but with all recorded neurons. Black = Monkey D, Gray = Monkey C. Black = Monkey D, Gray = Monkey C. $p < 10^{-5}$ for both correlation statistics.
(EPS)

**S7 Fig. Attribute encoding is modulated by gaze shifts in each subject.** Average beta coefficients, separated by monkey, from multiple regressions of fixation-related attribute values. Regressions were performed separately for the first and second fixation on trials with two or more pre-choice fixations. The same regressors, with variables defined by the first fixation, were used across both fixations. As in the main results, both animals separately showed that the average coefficients tended to be positive for the fixated attribute and negative for the like attribute of the other option (green and orange, respectively, at times after 0 on the first fixation and before 0 on the second fixation). Because monkeys tended to shift their gaze among like attributes, the second fixation was frequently directed to the attribute that was previously defined as the like attribute of the other option. Consistent with this, the average coefficients for that attribute became positive after the second fixation (orange), while the previously fixated attribute (green) became negative. Shading = ±sem.
(EPS)

## Acknowledgments

The authors thank Peter Rudebeck, Joey Charbonneau and Feng-Kuei Chiang for comments on the manuscript, and Fred Stoll for comments on the analyses.

## Author contributions

**Conceptualization:** Aster Q. Perkins, Erin L. Rich.

**Data curation:** Aster Q. Perkins.

**Formal analysis:** Aster Q. Perkins, Erin L. Rich.

**Funding acquisition:** Aster Q. Perkins, Erin L. Rich.

**Investigation:** Aster Q. Perkins.

**Methodology:** Aster Q. Perkins, Erin L. Rich.

**Software:** Aster Q. Perkins, Erin L. Rich.

**Visualization:** Erin L. Rich.

**Writing – original draft:** Erin L. Rich.

**Writing – review & editing:** Aster Q. Perkins, Erin L. Rich.

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
