## [Editor Report · Decision Letter 0]

Dear Dr Rich,

Thank you for submitting your manuscript entitled "Attention-dependent attribute comparisons underlie multi-attribute decision-making in orbitofrontal cortex" for consideration as a Research Article by PLOS Biology.

Your manuscript has now been evaluated by the PLOS Biology editorial staff, as well as by an academic editor with relevant expertise, and I am writing to let you know that we would like to send your submission out for external peer review.

Once your full submission is complete, your paper will undergo a series of checks in preparation for peer review. After your manuscript has passed the checks it will be sent out for review. To provide the metadata for your submission, please Login to Editorial Manager (https://www.editorialmanager.com/pbiology) within two working days, i.e. by Jan 20 2025 11:59PM.

Kind regards,

Taylor

Taylor Hart, PhD,

Associate Editor

PLOS Biology

thart@plos.org

---

## [Decision Letter · Decision Letter 1]

Dear Dr Rich,

Thank you for your patience while your manuscript "Attention-dependent attribute comparisons underlie multi-attribute decision-making in orbitofrontal cortex" was peer-reviewed at PLOS Biology. It has now been evaluated by the PLOS Biology editors, an Academic Editor with relevant expertise, and by two independent reviewers.

In light of the reviews, which you will find at the end of this email, we would like to invite you to revise the work to thoroughly address the reviewers' reports.

As you will see below, the reviewers are in agreement that the paper addresses an important question, and called the findings compelling and a significant advance. R1 was ready to accept the paper as is, but offered suggestions for strengthening the paper's message in the discussion. R2 raised several questions about the analyses and brought up alternative interpretations for some aspects of the findings, which will need to be addressed.

We now invite you to re-submit your manuscript after a Major Revision. The new version will need to include the additional analyses outlined by R2 in points 1A and 1B, as well as addressing all other concerns raised by the reviewers. We also encourage you to consider making textual changes in line with R1's suggestion.

Given the extent of revision needed, we cannot make a decision about publication until we have seen the revised manuscript and your response to the reviewers' comments. Your revised manuscript is likely to be sent for further evaluation by all or a subset of the reviewers.

**IMPORTANT - SUBMITTING YOUR REVISION**

*Re-submission Checklist*

*Published Peer Review*

*PLOS Data Policy*

*Blot and Gel Data Policy*

Sincerely,

Taylor

Taylor Hart, PhD,

Associate Editor

PLOS Biology

thart@plos.org

REVIEWS:

Reviewer #1: This paper reports data from monkeys making multi-attribute choices with neuronal recordings in OFC. The task involves sweetness and probability as independent dimensions. The key result is that (1) neurons encode the attended features not the integrated values, and (2) neurons encode the featural value difference. In addition, (3) the authors report clear attentional/ gaze effects. The authors conclude that OFC computes comparisons among attributes. The integration of dimensions into abstract value signals is a core issue - arguably THE core issue in neuroeconomics, and the OFC is a critical area for these questions. So the importance of the questions is above reproach. Overall, I was persuaded that the task (which is clever and powerful) is appropriate to the questions asked, and that the data are compelling and the analyses are done well.

In summary, this is an excellent paper that in my opinion could be published as it. Despite this, I nonetheless will provide some opinions, although I want to make clear these are suggestions rather than requirements for my approval, which I am already happy to give.

Most importantly, I felt that the core message of the paper is a bit muddled. This may be deliberate, since the findings are likely to be controversial and the authors may be eager to avoid controversy to achieve publication. If that is the case, I respect the decision, but in my view the authors are hiding their candle under a bushel.

To be specific, the authors findings are potentially quite revolutionary with respect to the role of OFC in economic choice. There is, to be sure, a moderate literature on this topic, including some of the authors' own work, and the authors do a very good job of citing this and summarizing it in the Discussion. In my view the standout paper in that set is Hunt et al., 2014. Other relevant ones include Tianming Yang's 2018 paper, and Vince McGinty's two papers. Those are all cited here, but, I felt that, together, they along with this paper, paint a portrait of OFC that is at odds with the generally accepted view of value integration, and that could be woven more clearly into the Introduction and Discussion to make a stronger case for importance. In particular, I left wanting a clearer answer to the question of "If OFC doesn't compute and compare abstract values then what does it do instead?" beyond simply a summary of all the results.

I have sometimes wondered whether what appear to be value comparison signals are instead normalized value signals (normalized relative to the other values on the menu). Those would look nearly identical in the data, but would have a very different meaning - they JUST encode the attended value (albeit normalized) and have no direct role in comparison.

SMALLER COMMENTS

In that vein, the authors' claim of possible multiple decision systems (L409) seems not really borne out by the data, since the authors seem to find contradictory, not complementary, results to these earlier studies. It seems the methods here are strong enough that the athors should have managed to replicate the earlier contradictory results, and their failure to do so raises the possibility the earlier ones were in error.

"Neurons that encode the integrated value of a choice should have non-zero regression coefficients for both attributes of the chosen option, simultaneously with the same sign" Since you require two things to pass threshold, your effective alpha is 0.025, I think. So to get an effective alpha of 0.05, you would have to use a real criterion of 0.1. In other words, the criteria are too strict. (Not sure about this, it depends on how the analyses took place).

"Instead, many neurons encoded the value of only one attribute, most frequently an attribute of the chosen option" but this involves drawing an inference from a failure to achieve significance, which is not really valid with null hypothesis statistical testing, right?

Finally, as a reviewer it would be easier to review with citations as names, not numbers.

Reviewer #2: This paper examines the coding and mechanisms of value comparison in orbitofrontal cortex (OFC) using single unit recording in macaque monkeys, and in particular: (i) whether value coding is dependent upon integrated value comparison, as is often assumed/argued, or whether it may instead depend upon local comparisons within-attribute; (ii) how these comparisons and coding mechanisms are modulated by gaze. To achieve this, two macaque monkeys perform a task in which they compare two options of differing sweetness/probability, indicated by bar heights. A clever manipulation of bar height to attribute value means that the authors can find behavioural evidence of the choices being consistent with a within-attribute comparison strategy, rather than (or, perhaps, in addition to) an integrated-value comparison strategy. Perhaps more importantly, the paper finds compelling evidence that, at least in the context of the present task, gaze-dependent value comparison occurs within attribute within the OFC (figure 5c and 6b).

The findings from this paper are generally very robust and have important implications for working models of how we think about gaze-dependent value coding in the OFC. They offer a significant advance on previous studies that have attempted to address related questions. There are, however, a few points that I think the authors may wish to address before publication.

1. The most striking finding - the one that really convinced me of the authors' central claim - is shown in figure 5c. There is clear evidence for anticorrelation in beta coefficients between similar attributes across the two options, but not in the other comparisons that are tested (e.g. fixated/paired attribute, which might be positive if integrated values were being encoded). There were, however, a couple of frames of reference that I was surprised the authors didn't consider in the paper, and I think would be worth testing, to really examine whether there is *any* encoding of integrated values (either at the single neuron or population level).

a. The first of these is looking at the relationship between sweetness encoding and probability encoding. It is possible that when examining an option, there is a population code for the two attributes specifically, but this is not in the better/worse frame of reference for the two attributes (cf. figure 1g), nor in the fixated/paired frame of reference for the attributes (cf. figure 5c left panel), but instead is consistently in the sweetness/probability frame of reference. (In other words, when fixating an option, the same neurons will code for sweetness no matter whether it is the fixated attribute or the paired attribute). Can the authors try an analysis equivalent to figure 5c, with fixated sweetness on the x-axis and fixated probability on the y-axis? If there is no correlation between these two variables, it might provide further evidence against the notion of integrated value coding in this task.

b. Another possibility - perhaps less likely, but worth testing nonetheless - is that the currently fixated attribute is compared against *both* attributes of the other option at the same time. This would be a similar analysis to figure 5c, but with fixated attribute beta on the x-axis and (other attribute, other option) on the y-axis. In other words, for the middle two panels of figure 5c, what happens if we swap y-axes across these two panels - do the negative correlations now disappear?

2. Although it is not central to the main claim of the paper, the test in figure 2b looks a little biased towards finding the result that is reported. It is easier to reject one null hypothesis (i.e. find 'one attribute only' value coding) than it is to simultaneously reject two null hypotheses (i.e. all the other possible coding schemes considered in this figure). This bias can actually be seen in the pre-stimulus baseline, where the proportion of neurons is green (false positives) is 0.05, as expected, whereas the proportion of neurons in the other colours is close to zero (as it is 0.05^2). The authors should think of a valid way to address this bias in the analysis.

3. Figure 2f/g: here it might also be interesting to consider the sweetness/probability frame of reference mentioned above. I'm also unsure what time-window is used here - can the authors clarify this (and also make sure the timewindow is mentioned for all other correlations that aren't sliding analyses in the paper? Apologies if I missed this somewhere).

4. What is the distribution of eye-movement latencies? This seems potentially important for interpreting some of the results in figure 2, which are locked to option presentation rather than fixation.

---

## [Decision Letter · Decision Letter 2]

Dear Dr Rich,

Thank you for your patience while we considered your revised manuscript "Attention-dependent attribute comparisons underlie multi-attribute decision-making in orbitofrontal cortex" for publication as a Research Article at PLOS Biology. This revised version of your manuscript has been evaluated by the PLOS Biology editors, the Academic Editor, and the original reviewers.

Based on the reviews which are found at the end of this letter, we are likely to accept this manuscript for publication, provided you satisfactorily address the remaining points raised by the reviewers.

IMPORTANT: Please also make sure to address the following data and other policy-related requests.

--TITLE:

We have two suggestions for the title. Are either of these options acceptable to you, and if so which do you prefer?

1. "Orbitofrontal cortex computes comparisons between attributes rather than integrated values"

Or

2. "Orbitofrontal cortex computes comparisons between attributes during multi-attribute decision making"

ETHICS STATEMENT:

-- Your article needs to include an Ethics Statement in the Methods section. Please add this (we see that you mention this in the "subjects" section of the methods) and ensure that it is consistent with the following guidelines:

-- Please include the full name of the IACUC/ethics committee that reviewed and approved the animal care and use protocol/permit/project license. Please also include an approval number.

-- Please include the specific national or international regulations/guidelines to which your animal care and use protocol adhered. Please note that institutional or accreditation organization guidelines (such as AAALAC) do not meet this requirement.

DATA POLICY:

-- We see that you have included a Data and Code Availability statement, where you say that you will make these items available before publication. Please now make these items available so that we can examine them before your article is formally accepted. Please ensure that these items are consistent with the following guidelines:

Fig. 1E

Fig. 3BCDEF

Fig. S1BCDEFGHIJ

Fig. S4AB

Fig. S5BCDEFG

CODE POLICY

SPECIES INDICATED IN THE ABSTRACT?

- We see that your abstract indicates that your study was done on monkeys. Can you please specify the species? Please note that per journal policy, the model system/species studied should be clearly stated in the abstract of your manuscript.

We expect to receive your revised manuscript within two weeks.

*Published Peer Review History*

*Press*

Sincerely,

Taylor

Taylor Hart, PhD,

Associate Editor

thart@plos.org

PLOS Biology

Reviewer remarks:

R1

I was positive about this paper before.

It is even stronger now.

I have no further concerns.

R2

The authors have addressed all the comments I raised in the initial round of reviews. It is a great paper and makes a strong contribution. I'd be inclined for them to include the weak negative correlation in response to my previous point 1b, either in the main panels or in the supplement, but I'm happy for the authors to make the final call on this.

R3

I thank the authors for answering my question in the response to reviews, but I would also like to see this information reported in the manuscript. It's true that Figures 4d-e and 6b show that the mean activity is inversely related, but these graphs do not by themselves show the inverse relationship at the single cell level. The inverse relationship at the single-cell level is, in my opinion, the most interesting and important result in this paper, and it should be fully reported. The fact that the cell-level correlations are different for all neurons vs. just the neurons in 5c is an important finding that helps further the understanding of the relevant neural mechanisms.

To be explicit, I think the manuscript should show the correlations in Fig. 5c for all neurons, preferably in a main or supplemental figure; if the authors choose to report these in text rather than a supplemental figure, they should include both the Pearson's R as well as some outlier-resistance correlation measure, such as Spearman's Rho. I think the manuscript should also report the correlations in Fig. 5c (blue and tan panels) for sweetness and probability separately; it's fine if these statistics are just for the subset of neurons in Fig. 5c.

A minor point, relevant to several figures: Figures with multiple sub-panels should have letter labels. For example, I would suggest that Fig. 5c use a separate letter for each of the four subpanels, i.e., 5c, 5d, 5e, 5f.

---

## [Editor Report · Decision Letter 3]

Dear Dr Rich,

Thank you for the submission of your revised Research Article "Orbitofrontal cortex computes gaze-dependent comparisons between attributes rather than integrated values" for publication in PLOS Biology. On behalf of my colleagues and the Academic Editor, Thorsten Kahnt, I am pleased to say that we can in principle accept your manuscript for publication, provided you address any remaining formatting and reporting issues. These will be detailed in an email you should receive within 2-3 business days from our colleagues in the journal operations team; no action is required from you until then. Please note that we will not be able to formally accept your manuscript and schedule it for publication until you have completed any requested changes.

PRESS

Sincerely, 

Taylor Hart, PhD,

Associate Editor

PLOS Biology

thart@plos.org